# Consistency Flow Matching: Defining Straight Flows with Velocity Consistency

## Abstract

Flow matching (FM) is a general framework for defining probability paths via Ordinary Differential Equations (ODEs) to transform between noise and data samples. Recent approaches attempt to straighten these flow trajectories to generate high-quality samples with fewer function evaluations, typically through iterative rectification methods or optimal transport solutions. In this paper, we introduce Consistency Flow Matching (Consistency-FM), a novel FM method that explicitly enforces self-consistency in the velocity field. Consistency-FM directly defines straight flows starting from different times to the same endpoint, imposing constraints on their velocity values. Additionally, we propose a multi-segment training approach for Consistency-FM to enhance expressiveness, achieving a better trade-off between sampling quality and speed. Extensive experiments demonstrate that our Consistency-FM significantly improves training efficiency by converging 4.4x faster than consistency models and 1.7x faster than rectified flow models while achieving better generation quality.

## 1 Introduction

In recent years, deep generative models have provide an attractive family of paradigms that can produce high-quality samples by modeling a data distribution, achieving promising results in many generative scenarios, such as image generation (Ho et al., 2020; Yang et al., 2023; 2024a;b). As a general and deterministic framework, Continuous Normalizing Flows (CNFs) (Chen et al., 2018) are capable of modeling arbitrary probability paths, specifically including the probability paths represented by diffusion processes (Song et al., 2021). To scale up the training of CNFs, many works propose efficient simulation-free approaches (Lipman et al., 2022; Albergo & Vanden-Eijnden, 2022; Liu et al., 2022) by parameterizing a vector field which flows from noise samples to data samples. Lipman et al. (2022) proposes Flow Matching (FM) to train CNFs based on constructing explicit conditional probability paths between the noise distribution and each data sample. Taking inspiration from denoising score matching (Song & Ermon, 2019), FM further shows that a per-example training objective can provide equivalent gradients without requiring explicit knowledge of the intractable target vector field, thus incorporating existing diffusion paths as special instances.

Straightness is one particularly-desired property of the trajectory induced by FM (Liu et al., 2022; 2023; Kornilov et al., 2024; Tong et al., 2023), because the straight path are not only the shortest path between two end points, but also can be exactly simulated without time discretization. To learn straight line paths which transport distribution $\pi_0$ to $\pi_1$, Liu et al. (2022) learn a rectified flow from data by turning an arbitrary coupling of $\pi_0$ and $\pi_1$ to a new deterministic coupling, and iteratively train new rectified flows with the data simulated from the previously obtained rectified flow. Some works resort to optimizing with an optimal transport plan by considering non-independent couplings of k-sample empirical distributions (Pooladian et al., 2023; Tong et al., 2023). For example, OT-CFM (Tong et al., 2023) attempts to approximate dynamic OT, creating simpler flows that are more stable to train and lead to faster inference.

However, despite their impressive generation quality, they still lack an effective trade-off between sampling quality and computational cost in straightening flows. To be more specifically, iterative rectification would suffer from accumulation error, and approximating an optimal transport plan in each training batch is computationally expensive. Therefore, a question naturally arises, *can one*

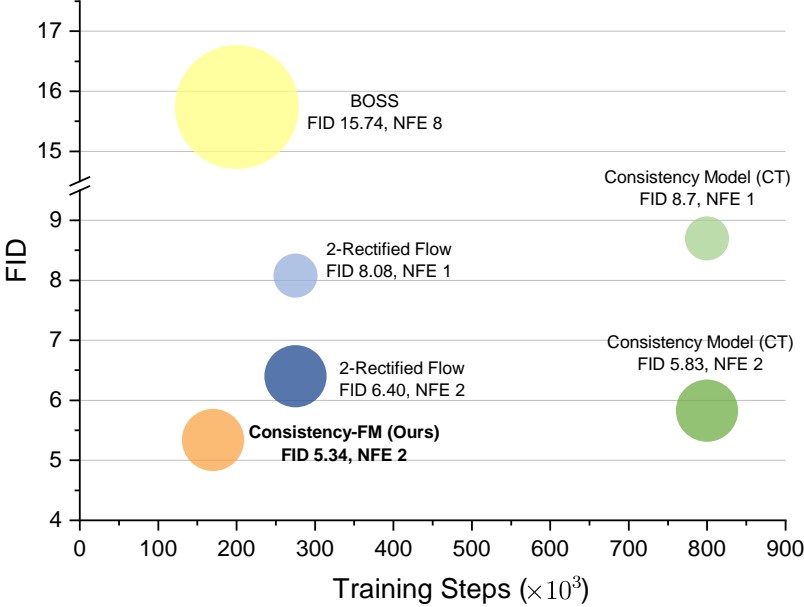

Figure 1: Comparison on CIFAR-10 dataset regarding the trade-off between generation quality and training efficiency. Our Consistency-FM demonstrates the best trade-off compared to consistency models (Song et al., 2023) and rectified flow models (Liu et al., 2022; Nguyen et al., 2024), **converging 4.4 times faster than consistency models and 1.7 times faster than rectified flow models** while achieving better generation quality

*learn an effective ODE model that maximally straightens the trajectories of probability flows without increasing training complexity?*

In this work, we propose a new fundamental FM method, namely **Consistency Flow Matching (Consistency-FM)**, to straighten the flows by explicitly enforcing self-consistency property in the velocity field. More specifically, Consistency-FM directly defines straight flows that start from different times to the same endpoint, and further constrains on their velocity values. To enhance the model expressiveness and enable better transporting between complex distributions, we resort to training Consistency-FM in a multi-segment approach, which constructs a piece-wise linear trajectory. Moreover, this flexible time-to-time jump allows Consistency-FM to perform distillation on pre-trained FM models for better trade-off between sampling speed and quality.

**Comparison with Consistency Models**   Consistency Models (CMs) (Song et al., 2023) learn a set of consistency functions that directly map noise to data. While CMs can generate sample with one NFE, they fail to provide a satisfying trade-off between generation quality and computational cost (Kim et al., 2023). Moreover, enforcing consistency property at arbitrary points is redundant and potentially slows down the training process. In contrast, our Consistency-FM enforces the consistency property over the space of velocity field instead of sample space, which can be viewed as a high-level regularization for straightening ODE trajectory. While CMs are able to learn consistency functions in a general form, Consistency-FM parameterizes the consistency functions as straight flows, which enables faster training convergence without the need for approximating the entire probability path.

**Main Contributions**   We summarize our contributions as follows: (i) We propose a new fundamental class of FM models that explicitly enforces the self-consistency in the space of velocity field instead of sample space. (ii) We conduct sufficient theoretical analysis for our proposed Consistency-FM, and enhance its expressiveness with multi-segment optimization. (iii) Extensive experiments on three classical image datasets demonstrate the superior generation quality and training efficiency of our Consistency-FM (e.g., 4.4 times and 1.7 times faster than consistency model and rectified flow). Further text-to-image experiments sufficiently prove our effectiveness and generalization ability.

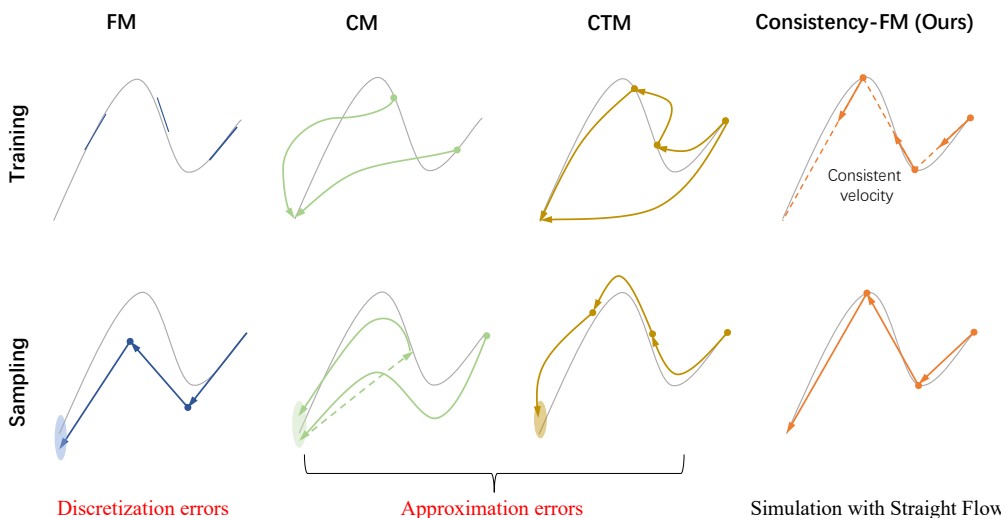

Figure 2: Training and sampling comparisons between flow matching (FM) (Lipman et al., 2022), consistency model (CM) (Song et al., 2023) and consistency trajectory model (CTM) (Kim et al., 2023) and our Consistency-FM. While previous methods can cause discretization errors or approximation errors, Consistency-FM mitigates these issues by defining straight flows in simulation.

## 2 RELATED WORK AND DISCUSSIONS

**Flow Matching for Generative Modeling**  Flow Matching (FM) aims to (implicitly) learn a vector field $\{v_t\}_{t \in [0,1]}$, which generates an ODE that admits to the desired probability path $\{p_t\}_{t \in [0,1]}$ (Lipman et al., 2022). The training of FM does not require any computational challenging simulation, as it directly estimate the vector field using a regression objective which can be efficiently estimated (Lipman et al., 2022). By the construction of FM, it allows general trajectory of ODE and probability path, thus many effort have been dedicated to design better trajectory with certain properties(Pooladian et al., 2023; Tong et al., 2023; Klein et al., 2023; Stark et al., 2024; Campbell et al., 2024). One particularly desired property is the straightness of the trajectory, as a straight trajectory can be efficiently simulated with few steps of Euler integration. Concurrent works Multi-sample FM (Pooladian et al., 2023) and Minibatch OT (Tong et al., 2023) propose to generalize the independent coupling of data distribution $p_0(x_0)$ and prior distribution $p_1(x_1)$ to optimal transport coupling plan $\pi(x_0, x_1)$. Under the optimal transport plan, the learned trajectory of ODE will tend to be straight. However, their methods require constructing the approximated optimal transport plan in each training batch, which is computationally prohibitive.

Rectified Flow (Liu et al., 2022; 2023) can be viewed as a FM with specific trajectory. Rectified Flow proposes to rewire and straighten the trajectory by iterative distillation, which requires multiple round of training and may suffer from accumulation error. A recent work, Optimal FM (Kornilov et al., 2024) proposes to directly learn the optimal transport map from $p_1$ to $p_0$ and use it to calculate the vector field and straight trajectory. However, computing the optimal transport map in high dimension is a challenging task (Makkuva et al., 2020), and Optimal FM (Kornilov et al., 2024) only provides experiments on toy datasets. In this paper, we propose to straighten the trajectory in a more flexible and effective approach by enforcing the self-consistency property in the velocity field.

**Learning Efficient Generative Models**  GANs (Arjovsky et al., 2017; Goodfellow et al., 2014), VAEs (Kingma & Welling, 2013), Diffusion Models (Song et al., 2020b;a; Ho et al., 2020) and Normalizing Flows (Rezende & Mohamed, 2015; Dinh et al., 2016) have been four classical deep generative models. Among them, GANs and VAEs are efficient one-step models. However, GANs usually suffer from the training instability and mode collapse issues, and VAEs may struggle to generate high-quality examples. Therefore, recent works begin to utilize diffusion models and continuous normalizing flows (Chen et al., 2018) for better training stability and high-fidelity generation, which are based on a sequence of expressive transformations for generative sampling.

To achieve a better trade-off between sampling quality and speed, many efforts have been made to accelerate diffusion models, either by modifying the diffusion process (Song et al., 2020a; Bao et al., 2021; Dockhorn et al., 2021; Xiao et al., 2021; Yang et al., 2024b; Wang et al., 2024), with an efficient ODE solver (Lu et al., 2022; Dockhorn et al., 2022; Zheng et al., 2023), or performing distillation between pre-trained diffusion models and their more efficient versions (e.g., with less sampling steps) (Salimans & Ho, 2022; Liu et al., 2022; Luo et al., 2024; Luo, 2023). However, most distillation methods require multiple training rounds and are susceptible to accumulation errors. Recent Consistency Models (Song et al., 2023; Song & Dhariwal, 2024) distill the entire sampling process of diffusion model into one-step generation, while maintaining good sample quality. Consistency Trajectory Models (CTMs)(Kim et al., 2023) further mitigate the issues about the accumulated errors in multi-step sampling. However, these methods must learn to integrate the full ODE integral, which are difficult to learn when it jumps between modes of the target distribution. In this paper, we propose a new concept of *velocity consistency* with defined straight probability flows, achieving most competitive results on both one- and multi-step generation.

## 3 CONSISTENCY FLOW MATCHING

### 3.1 PRELIMINARIES ON FLOW MATCHING

Let $\mathcal{R}^d$ denote the data space with data point $x_0 \in \mathcal{R}^d$, FMs aim to the learn a vector field $v_\theta(t, x) : [0, 1] \times \mathcal{R}^d \to \mathcal{R}^d$, such that the solution of the following ODE can transport noise $x_0 \sim p_0$ to data $x_1 \sim p_1$:

$$\begin{cases} \dfrac{d\gamma_x(t)}{dt} = v_\theta(t, \gamma_x(t)), \\ \gamma_x(0) = x \end{cases} \tag{1}$$

The solution of Eq. (1) is denoted by $\gamma_x(\cdot)$, which is also called a flow, describing the trajectory of the ODE from starting point $x$. Given the ground truth vector field $u(t, x)$ that generates probability path $p_t$ under the two marginal constraints that $p_{t=0} = p_0$ and $p_{t=1} = p_1$, FMs seek to optimize the simple regression objective

$$E_{t,p_t} ||v_\theta(t, x_t) - u(t, x_t)||_2^2 \tag{2}$$

However, it is computational intractable to find such $u$, since $u$ and $p_t$ are governed by the following continuity equation (Villani et al., 2009):

$$\partial_t p_t(x) = -\nabla \cdot (u(t, x)p_t(x)) \tag{3}$$

Instead of directly optimizing Eq. (2), Conditional Flow Matching (Lipman et al., 2022) regress $v_\theta(t, x)$ on the conditional vector filed $u(t, x_t|x_1)$ and probability path $p_t(x_t|x_1)$ :

$$E_{t,q(x_1)} E_{p_t(x_t|x_1)} ||v_\theta(t, x_t) - u(t, x_t|x_1)||_2^2 \tag{4}$$

Two objectives Eq. (2) and Eq. (4) share the same gradient with respect to $\theta$, while Eq. (4) can be efficiently estimated as long as the conditional pair $u(t, x_t|x_1), p_t(x_t|x_1)$ is tractable. Note that recovering the marginal vector field and probability path from the conditioned one remains a complex challenge (Lipman et al., 2022).

### 3.2 DEFINING STRAIGHT FLOWS WITH CONSISTENT VELOCITY

**Motivation** Recent FM methods for learning straight flows typically necessitate the approximation the probability path $p_t$ and its marginal distributions $p_0$ and $p_1$ (Liu et al., 2022; 2023; Pooladian et al., 2023; Lee et al., 2023) , which are computational intensive and introduce additional approximation error. To address these challenges, we introduce Consistency-FM, a general method to efficiently learn straight flows without the need for approximating the entire probability path.

A straightforward approach to learn straight flows is to identify a consistent ground truth vector field and then use objective in Eq. (2) for training. The definition of consistent velocity is $v(t, \gamma_x(t)) = v(0, x)$, indicating the velocity along the solution of Eq. (1) remains constant. However, due to the intractability of original data distribution, it is also intractable to find such a vector field, or to design a conditional vector field such that the corresponding marginal velocity is consistent (Lipman et al., 2022; Pooladian et al., 2023).

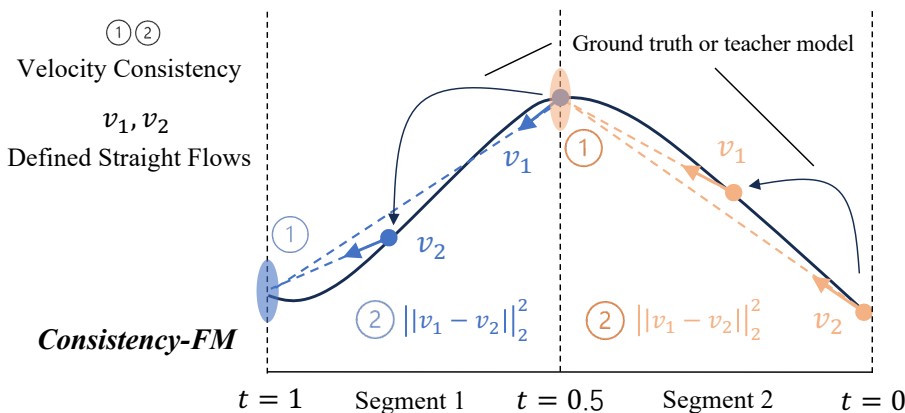

Figure 3: Ilustration of training our consistency-FM.

Instead of directly regressing on the ground truth vector field, Consistency-FM directly defines straight flows with consistent velocity that start from different times to the same endpoint. Specifically, we have the following lemma (prove in Appendix A.1):

**Lemma 1.** *Assuming the vector field is Lipschitz with respect to $x$ and uniform in $t$, and are differentiable in both input, then these two conditions are equivalent:*

*Condition 1.* $\quad v(t, \gamma_x(t)) = v(s, \gamma_x(s)), \quad \forall t, s \in [0, 1]$

*Condition 2.* $\quad \gamma_x(t) + (1 - t) * v(t, \gamma_x(t)) = \gamma_x(s) + (1 - s) * v(s, \gamma_x(s)), \quad \forall t, s \in [0, 1],$ (5)

where $\gamma_x(t)$ represents the solution of Eq. (1) at time $t$. *Condition 2* specifies that starting from an arbitrary time $t$ with data point $\gamma_x(t)$, and moving in the direction of current velocity for a duration of $1 - t$, the resulting data will be consistent and independent with respect to $t$.

**Velocity Consistency Loss** While *Condition 1* directly constraints the vector field to be consistent, learning vector fields that only satisfy *Condition 1* may lead to trivial solutions. On the other hand, *Condition 2* ensures the consistency of the vector field from a trajectory viewpoint, offering a way to directly define straight flows. Motivated by this, Consistency-FM learns a consistency vector field to satisfy both conditions:

$$\mathcal{L}_\theta = E_{t \sim \mathcal{U}} E_{x_t, x_{t+\Delta t}} ||f_\theta(t, x_t) - f_{\theta^-}(t + \Delta t, x_{t+\Delta t})||_2^2 + \alpha ||v_\theta(t, x_t) - v_{\theta^-}(t + \Delta t, x_{t+\Delta t})||_2^2,$$

$$f_\theta(t, x_t) = x_t + (1 - t) * v_\theta(t, x_t),$$

(6)

where $\mathcal{U}$ is the uniform distribution on $[0, 1 - \Delta t]$, $\alpha$ is a positive scalar, $\Delta t$ denotes a time interval which is a small and positive scalar. $\theta^-$ denotes the running average of past values of $\theta$ using exponential moving average (EMA), $x_t$ and $x_{t+\Delta t}$ follows a pre-defined distribution which can be efficiently sampled, for example, VP-SDE (Ho et al., 2020) or OT path (Lipman et al., 2022). Note that by setting $t = 1$, *Condition 2* implies that $\gamma_x(t) + (1 - t) * v(t, \gamma_x(t)) = \gamma_x(1) \sim p_1$, and thus training with $\mathcal{L}_\theta$ can not only regularize the velocity but also learn the data distribution. Furthermore, if *Condition 2* is met, then the straight flows $\gamma_x(t) + (1 - t) * v(t, \gamma_x(t))$ can directly predict $x_1$ from each time point $t$ (Song et al., 2023).

Below we provide a theoretical justification for the objective based on asymptotic analysis (proof in Appendix A.3).

**Theorem 1.** *Consider no exponential moving average, i.e., $\theta^- = \theta$. Assume there exists ground truth velocity field $u_t$ that generates $p_t$ and satisfies the continuity Eq. (3). Furthermore we assume $v_\theta$ is bounded and twice continuously differentiable with bounded first and second derivatives, the ground truth velocity $u_t$ is bounded. Then we have:*

$$E||f_\theta(t, x_t) - f_\theta(t + \Delta t, x_{t+\Delta t})||_2^2 = (\Delta t)^2 E||v_\theta(t, x_t) - u(t, x_t) - (1 - t)(\partial_t v_\theta + u \cdot \nabla_x v_\theta)||_2^2 + o((\Delta t)^2)$$

(7)

**Remark 1.** The objective in Eq. (7),

$$E||v_\theta(t, x_t) - u(t, x_t) - (1-t)(\partial_t v_\theta + u \cdot \nabla_x v_\theta)||_2^2,$$

can be seen as striking a balance between exact velocity estimation and adhering to consistent velocity constraints. On the one hand, the objective aims to minimize the discrepancy between learned and ground truth velocity $v_\theta(t, x_t) - u(t, x_t)$, aligning with the goal of FM-based methods (Lipman et al., 2022). On the other hand, it also considers the consistency of the velocity. By Lemma 2 in the Appendix, $\partial_t v_\theta + u \cdot \nabla_x v_\theta$ serves as a constraint for velocity consistency, which measures the changes of the velocity after taking a infinitesimal step along the direction of ground truth velocity. Given the ground truth velocity may not be consistent, this objective provides a trade-off between the sampling quality and computational cost with straight flow.

### 3.3 MULTI-SEGMENT CONSISTENCY-FM

To enhance the expressiveness of Consistency-FM for transporting distributions in general probability path, we introduce Multi-Segment Consistency-FM. This approach relaxes the requirement for consistent velocity throughout the flow, allowing for more flexible adaptations to diverse distribution characteristics. Multi-Segment Consistency-FM divides the time interval into equal segments, learning a consistent vector field $v_\theta^i$ within each segment. After recombining these segments, it constructs a piece-wise linear trajectory to transport noise to data distribution. Specifically, given a segment number $K$, the time interval $[0, 1]$ is divided with $[0, 1] = \Sigma_{i=0}^{K-1}[i/K, (i+1)/K]$. Then the training objective is defined as

$$\mathcal{L}_\theta = E_{t \sim \mathcal{U}^i} \lambda^i E_{x_t, x_{t+\Delta t}} ||f_\theta^i(t, x_t) - f_{\theta^-}^i(t+\Delta t, x_{t+\Delta t})||_2^2 + \alpha ||v_\theta^i(t, x_t) - v_{\theta^-}^i(t+\Delta t, x_{t+\Delta t})||_2^2,$$

$$f_\theta^i(t, x_t) = x_t + ((i+1)/k - t) * v_\theta^i(t, x_t), \tag{8}$$

where $i$ denotes the $i^{th}$ segment, $\mathcal{U}^i$ is the uniform distribution on $[i/K, (i+1)/K - \Delta t]$, $x_t, x_{t+\Delta t}$ follow a pre-defined distribution, $\Delta t$ is a small and positive constant . $v_\theta^i(t, x_t)$ are the flow and the consistent vector field in segment $i$, respectively. $\lambda^i$ is a positive weighting scalar for different segment, as vector field in the middle of $[0, 1]$ is more difficult to train (Esser et al., 2024).

Below we provide a theoretical justification for multi-segment training. First, we analyse the optimal solution for objective Eq. (8) and provide an explicit formula for the estimation error in Multi-Segment Consistency-FM (proof in Appendix Appendix A.4):

**Theorem 2.** *Consider training consistency-FM on segment $i$ which is defined in time interval $[S, T]$. Assume there exists ground truth velocity field $u_t$ that generates $p_t$ and satisfies the continuity equation Eq. (3), let $v^*(t, x)$ denote the oracle consistent velocity such that*

$$x_T = x_t + (T-t)v^*(t, x_t). \tag{9}$$

*Then the learned $v_\theta^i(t, x_t)$ that minimize Eq. (8) in segment $i$ at time $t \in [S, T - \Delta t]$ has the following error:*

$$v_\theta^i(t, x_t) - v^*(t, x_t) = \frac{\alpha}{(T-t)^2 + \alpha}(v^*(t+\Delta t, x_{t+\Delta t}) - v^*(t, x_t))$$

$$+ \frac{(T-t-\Delta t)(T-t) + \alpha}{(T-t)^2 + \alpha}(v_\theta^i(t+\Delta t, x_{t+\Delta t}) - v^*(t+\Delta t, x_{t+\Delta t})) \tag{10}$$

**Remark 2.** The mismatch between the learned velocity $v_\theta^i(t, x_t)$ and oracle velocity $v^*(t, x)$ is composed of two parts. The first part is the inconsistency of the oracle $v^*(t+\Delta t, x_{t+\Delta t}) - v^*(t, x_t)$, which is due to the fact that the ground truth velocity $u(t, x_t)$ might not be consistent. If $u(t, x_t)$ is consistent within the time interval $[S, T]$, then the oracle velocity is the ground truth velocity $v^*(t, x) = u(t, x_t)$ and $v^*(t, x)$ is consistent, and thus the error in the first part will vanish. The second part is the accumulated error from prior time step $t + \Delta t$, and by induction, we can deduce that this part will also vanish if the $u(t, x_t)$ is consistent. As a result, Consistency-FM can learn the ground truth velocity with objective Eq. (4) on any time interval $[S, T]$ where the ground truth velocity is consistent.

**Corollary 2.1.** *Consider training consistency-FM on segment $i$ which is defined in time interval $[S, T]$. Assume there exists ground truth velocity field $u_t$ that generates $p_t$ and satisfies the continuity equation Eq. (3). If the ground truth velocity $u$ is consistent within $[S, T]$, then Consistency-FM can learn the ground truth velocity, i.e., the learned $v_\theta(t, x_t) = u(t, x_t)$ almost everywhere.*

Table 1: Performance comparisons on CIFAR-10.

| Method | NFE ($\downarrow$) | FID ($\downarrow$) |
|---|---|---|
| Score SDE (Song et al., 2020b) | 2000 | 2.20 |
| DDPM (Ho et al., 2020) | 1000 | 3.17 |
| LSGM (Vahdat et al., 2021) | 147 | 2.10 |
| PFGM (Xu et al., 2022) | 110 | 2.35 |
| EDM (Karras et al., 2022) | 35 | **2.04** |
| **Direct Training** | | |
| 1-Rectified Flow (Liu et al., 2022) | 1 | 378 |
| Glow (Kingma & Dhariwal, 2018) | 1 | 48.9 |
| Residual Flow (Chen et al., 2019) | 1 | 46.4 |
| GLFlow (Xiao et al., 2019) | 1 | 44.6 |
| DenseFlow (Grcić et al., 2021) | 1 | 34.9 |
| Consistency Training (Song et al., 2023) | 2 | 5.83 |
| **Consistency-FM** | 2 | **5.34** |
| iCT-deep (Song & Dhariwal, 2024) | 2 | 2.24 |
| TRACT (Berthelot et al., 2023) | 2 | 3.32 |
| MultiStep-CT (Heek et al., 2024) | 2 | - |
| CTM (GAN loss) (Kim et al., 2023) | 2 | 1.93 |
| **Consistency-FM (GAN loss)** | 2 | **1.75** |
| **Diffusion Models - Distillation Sampling** | | |
| Consistency Distillation (Song et al., 2023) | 2 | 2.93 |
| CTM (GAN loss) (Song et al., 2023) | 2 | 1.87 |
| **Consistency-FM (GAN loss)** | 2 | **1.69** |

Table 2: Performance comparisons on ImageNet $64 \times 64$.

| Model | NFE | FID$\downarrow$ | Rec$\uparrow$ |
|---|---|---|---|
| Validation Data | | 1.41 | 0.67 |
| ADM (Dhariwal & Nichol, 2021) | 250 | 2.07 | 0.63 |
| EDM (Karras et al., 2022) | 79 | 2.44 | **0.67** |
| BigGAN-deep (Brock et al., 2018) | 1 | 4.06 | 0.48 |
| StyleGAN-XL (Sauer et al., 2022) | 1 | 2.09 | 0.52 |
| **Diffusion Models – Distillation Sampling** | | | |
| PD (Salimans & Ho, 2022) | 1 | 15.39 | 0.62 |
| BOOT (Gu et al., 2023) | 1 | 16.3 | 0.36 |
| TRACT (Berthelot et al., 2023) | 1 | 7.43 | - |
| CD (Song et al., 2023) | 1 | 6.20 | 0.63 |
| **Consistency-FM** | 1 | 5.43 | 0.61 |
| PD (Salimans & Ho, 2022) | 2 | 8.95 | 0.65 |
| TRACT (Berthelot et al., 2023) | 2 | 4.97 | - |
| CD (Song et al., 2023) | 2 | 4.70 | 0.64 |
| iCT-deep (Song & Dhariwal, 2024) | 2 | 2.77 | 0.62 |
| MultiStep-CD (Heek et al., 2024) | 2 | 1.90 | - |
| CTM (GAN loss) | 2 | 1.73 | 0.57 |
| **Consistency-FM (GAN loss)** | 2 | **1.62** | 0.56 |
| **Direct Training** | | | |
| Consistency Training (Song et al., 2023) | 2 | 11.1 | 0.56 |
| **Consistency-FM (GAN loss)** | 2 | **9.58** | 0.54 |

**Distillation with Consistency-FM** Consistency-FM can also be trained with pre-trained FMs. For distillation from a pre-trained FM $u_\phi(t, x_t)$, the consistency distillation loss for Consistency-FM is defined as

$$\mathcal{L}_{\theta,\phi} = E_{t \sim \mathcal{U}} E_{x_t} ||f_\theta(t, x_t) - f_{\theta^-}(t + \Delta t, \hat{x}^\phi_{t+\Delta t})||_2^2 + \alpha ||v_\theta(t, x_t) - v_{\theta^-}(t + \Delta t, \hat{x}^\phi_{t+\Delta t})||_2^2,$$

$$f_\theta(t, x_t) = x_t + (1 - t) * v_\theta(t, x_t),$$

$$\hat{x}^\phi_{t+\Delta t} = x_t + \Delta t * u_\phi(t, x_t),$$

(11)

where $\mathcal{U}[0, 1 - \Delta t]$ is the uniform distribution, $u_\phi(t, x)$ is the pre-trained FM, $x_t$ follows the distribution from which $u_\phi$ is trained, $\hat{x}^\phi_{t+\Delta t}$ is the one-step prediction using pre-trained model. For distillation from a pre-trained FMs, we set the segment number $K = 1$, as evidences show that the flows in pre-trained FMs are relatively straight (Liu et al., 2022; Pooladian et al., 2023).

**Sampling with Consistency-FM** Consistency-FM facilitates both one-step and multi-step generation. With a well-trained Consistency FM $v_\theta(\cdot, \cdot)$, we can generate sample by sampling from prior distribution $x_0 = p_0$ and then evaluating the model to transport the data through $k$ segments:

$$x_{i/k} = x_{(i-1)/k} + 1/k * v_\theta^i((i-1)/k, x_{(i-1)/k}), i = 1, 2, \ldots k - 1 \quad (12)$$

This approach offers a versatile framework that facilitates a balanced trade-off between sample quality and sampling efficiency.

## 4 EXPERIMENTS

**Implementation Details** We evaluate our Consistency-FM on both unconditional and conditional image generation tasks. Following previous methods (Liu et al., 2022; 2023; Yan et al., 2024), we use the CIFAR-10 (Alex, 2009), ImageNet (Deng et al., 2009), CelebA-HQ (Karras et al., 2017) and AFHQ-Cat (Choi et al., 2020), and MS-COCO dataset (Lin et al., 2014) for comprehensive evaluations, and we calculate the FID and CLIP scores for measurements. In distillation experiments, we follow previous methods (Liu et al., 2023; Yan et al., 2024) to use Stable Diffusion v1.5 (Rombach et al., 2022) as the backbone model. We set $\Delta t = 0.001$ in all experiments. Multi-segment consistency-FM is applied only for experiments where the number of function evaluations (NFE) is greater than 1. And the NFE is equivalent to the number of segments for the multi-segment Consistency-FM. We employ Consistency-FM based distillation training with a batch size of 128 for 200,000 iterations, consuming a total of 25 A100 GPU days. In comparison, the distillation process of InstaFlow (Liu et al., 2023) requires 110 A100 GPU days. Thus our training cost is only 23% of that of InstaFlow's distillation process. More experimental settings can be found in Table 5 of the Appendix B, and we also provide our code in the supplementary materials.

Table 3: Comparing Consistency-FM with previous flow matching models.

| Method | AFHQ-Cat 256 × 256 | | CelebA-HQ 256 × 256 | |
|---|---|---|---|---|
| | NFE ($\downarrow$) | FID ($\downarrow$) | NFE ($\downarrow$) | FID ($\downarrow$) |
| Rectified Flow (Liu et al., 2022) | 8 | 57.0 | 8 | 109.4 |
| Rectified Flow + Bellman Sampling (Nguyen et al., 2024) | 8 | 33.9 | 8 | 49.8 |
| Rectified Flow (Liu et al., 2022) | 6 | 61.5 | 6 | 127.0 |
| Rectified Flow + Bellman Sampling (Nguyen et al., 2024) | 6 | 36.2 | 6 | 72.5 |
| **Consistency-FM** | 6 | **22.5** | 6 | **36.4** |

**Baseline Methods**  To demonstrate the effectiveness of our Consistency-FM, we follow previous work (Song et al., 2023) and compare Consistency-FM with some representative diffusion models and flow models, such as Consistency Model (Song et al., 2023) and Rectified Flow (Liu et al., 2022). In the experiments on AFHQ-Cat and CelebA-HQ datasets, we also add recent Bellman Sampling (Nguyen et al., 2024) for flow matching models as the baseline. In distillation experiments, we mainly compare our method with InstaFlow (Liu et al., 2023) and PeRFlow (Yan et al., 2024), which are all flow matching based methods.

## 4.1 CONSISTENCY-FM BEATS RECTIFIED FLOW AND CONSISTENCY MODEL

As demonstrated in Table 1 and Table 2, on CIFAR-10 and ImageNet dataset, our Consistency-FM not only surpasses representative efficient generative models like Consistency Model and Rectified Flow, but also outperforms the powerful CTM (Kim et al., 2023) equipped with the same GAN loss as CTM. Notably, in Fig. 1, our Consistency-FM significantly advances training efficiency, **converging 4.4 times faster than consistency model and 1.7 times faster than rectified flow** while achieving superior sampling quality. These evaluation results sufficiently show that our Consistency-FM provides a more efficient way to model data distribution, proving the efficacy of our proposed learning paradigm of velocity consistency for FM models.

Table 3 shows the quantitative result of FM models and our Consistency-FM on high-resolution (256 × 256) image generation, including AFHQ-Cat and CelebA-HQ. We can observe that our Consistency-FM also outperform existing SOTA FM methods like rectified flow and rectified flow + Bellman sampling (Nguyen et al., 2024) by a significant margin with same NFEs. Furthermore, compared to CIFAR-10, Consistency-FM shows a greater improvement in generating high-resolution images. This phenomenon demonstrates that our Consistency-FM can potentially learn straighter flows for modeling more complex data distribution, enabling faster and better sampling.

## 4.2 DISTILLATION FOR ACCELERATING TEXT-TO-IMAGE GENERATION

We perform evaluations on few-step text-to-image generation to demonstrate our generalization ability. As shown in Table 7 and Table 6, our Consistency-FM demonstrates best trade-off between generation quality and sampling efficiency. More specifically, compared to previous FM-based acceleration methods, our Consistency-FM can achieve higher FID and CLIP scores, which means our velocity consistency loss can enable the model to better fit complex data distribution and improve its controllability. The results in the Table 6 of Appendix B further show that our Consistency-FM can also surpass previous diffusion-based acceleration methods, such as DMD (Yin et al., 2024). These surprising results prove the effectiveness and efficiency of our Consistency-FM, demonstrating great potential more challenging conditional generation tasks.

Table 4: Comparison of FID on MS COCO 2017 following the evaluation setup in Liu et al. (2023).

| Method | Inference Time | FID-5k | CLIP |
|---|---|---|---|
| PD-SD (1 step) (Salimans & Ho, 2022) | 0.09s | 37.2 | 0.275 |
| PD-SD (2 step) (Salimans & Ho, 2022) | 0.13s | 26.0 | 0.297 |
| PD-SD (4 step) (Salimans & Ho, 2022) | 0.21s | 26.4 | 0.300 |
| PeRFlow (4 step) (Yan et al., 2024) | 0.21s | 23.0 | 0.294 |
| 2-RF (2 step) (Liu et al., 2022) | 0.13s | 31.3 | 0.296 |
| 2-RF (1 step) (Liu et al., 2022) | 0.09s | 47.0 | 0.271 |
| InstaFlow (Liu et al., 2023) | 0.09s | 23.4 | 0.304 |
| **Consistency-FM** | 0.09s | **22.7** | **0.307** |

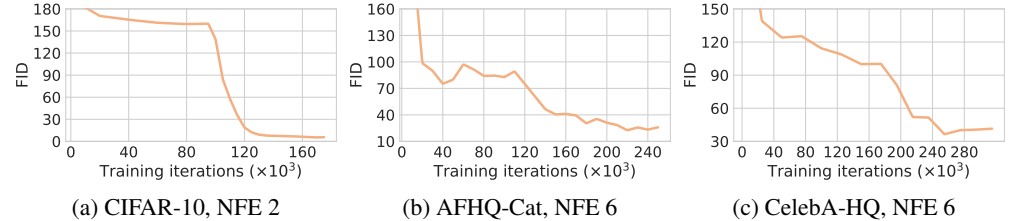

(a) CIFAR-10, NFE 2    (b) AFHQ-Cat, NFE 6    (c) CelebA-HQ, NFE 6

Figure 4: Demonstration of training convergence on three datasets.

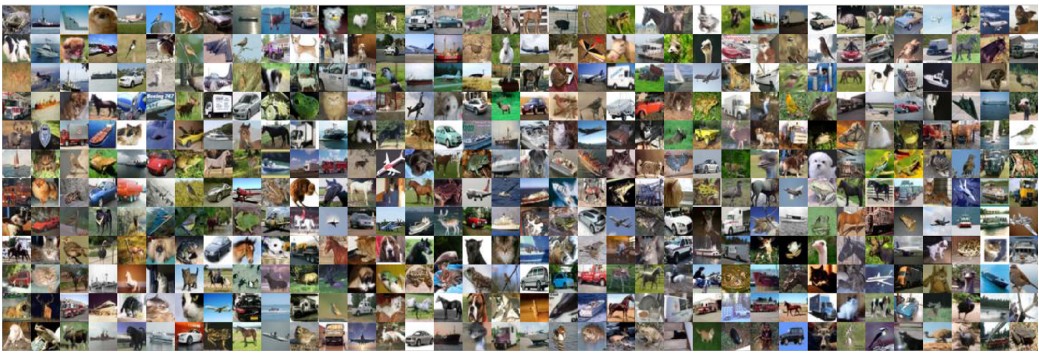

Real Images in CIFAR-10

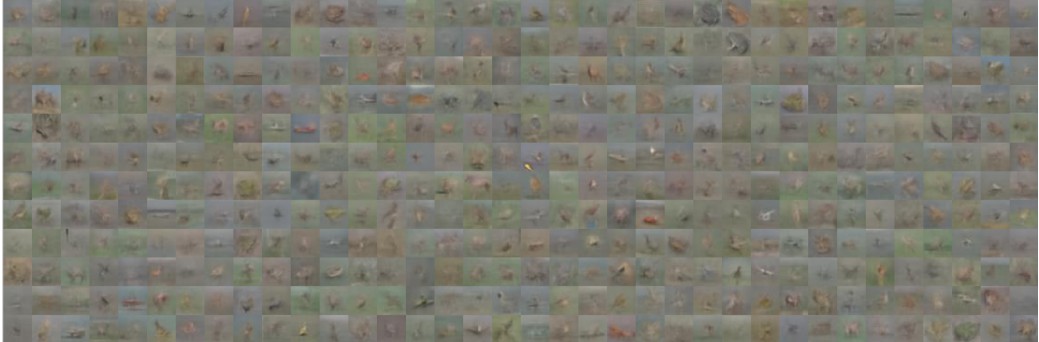

1-Rectified Flow, NFE=2

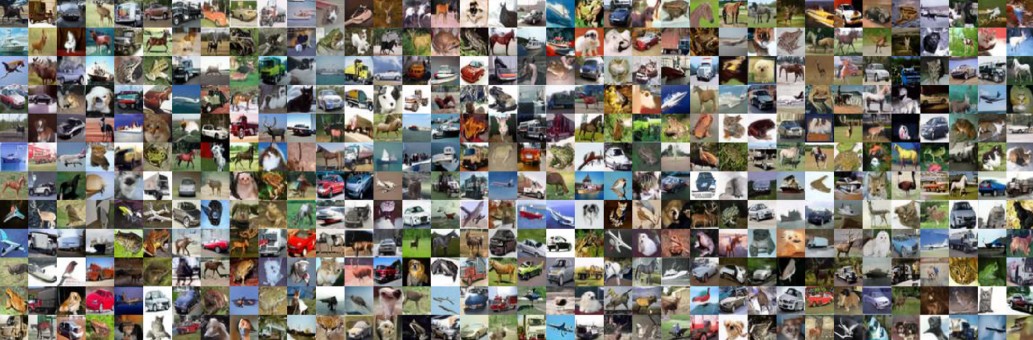

Consistency-FM (Ours), NFE=2

Figure 5: Sampling comparison between Rectified Flow (Liu et al., 2022) and our Consistency-FM.

### 4.3 QUALITATIVE ANALYSIS

We provide three convergence processes of training our Consistency-FM in Fig. 4. We observe that Consistency-FM converges faster on CIFAR-10 than on AFHQ-Cat and CelebA-HQ because

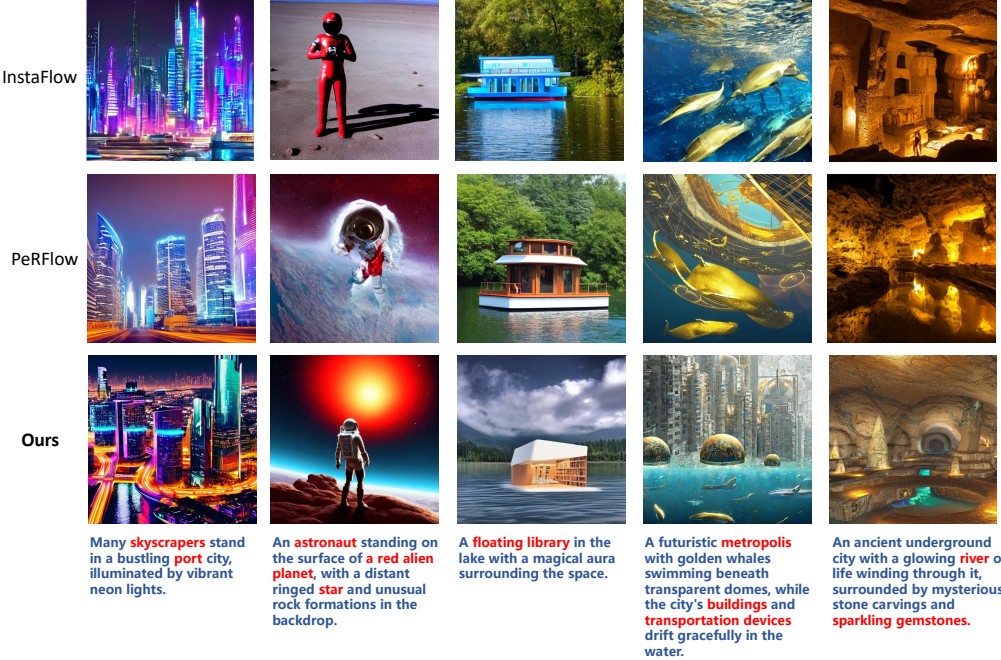

Figure 6: Qualitative comparison on complex text-to-image generation between InstaFlow (Liu et al., 2023), PeRFlow (Yan et al., 2024) and our Consistency-FM. We use red texts to denote critical semantics, and our Consistency-FM demonstrates superior generation qualities.

the latter two high-resolution datasets are more complex to model their data distributions. Overall, Consistency-FM consistently converges fast, proving the efficacy of defining straight flows for generative modeling. Additionally, we qualitatively compare our method with rectified flow in Fig. 5. From the generation results, we can observe that our Consistency-FM is capable of generating more realistic images than rectified flow with the same NFEs, revealing our Consistency-FM models data distribution more effectively.

In Fig. 6, We present a comparison of the images generated by InstaFlow (Liu et al., 2023) and PeRFlow (Yan et al., 2024) across different complex text-to-image generation scenarios. From the results, we can find that our Consistency-FM consistently outperforms previous methods, especially in complex semantics. We attribute this to our velocity consistency functions and multi-segment strategies, greatly enhancing the expressiveness and controllability of our Consistency-FM.

## 5 CONCLUSION AND FUTURE WORK

In this paper, we introduce a new fundamental class of FM models, namely Consistency Flow Matching (Consistency-FM), to explicitly enforces self-consistency in the velocity field. Consistency-FM directly defines straight flows starting from different times to the same endpoint, and is optimized by a multi-segment training approach for enhancing expressiveness. This work theoretically and empirically presents our new fundamental flow matching model. Extensive experiments demonstrate that our Consistency-FM significantly improves training efficiency by converging 4.4x faster than consistency models and 1.7x faster than rectified flow models while achieving better generation quality. Based on our Consistency-FM, we here propose two potential future research directions:

- **Compositional Text-to-Image Generation**: Our Consistency-FM demonstrates superior text-to-image results, especially in conveying complex textual semantics. By applying Consistency-FM to existing compositional T2I models (Yang et al., 2024a), we believe this would significantly improve both efficiency and quality of existing methods.
- **Efficient Video Generation**: Training video diffusion models is a time-consuming procedure (Blattmann et al., 2023; Tian et al., 2024), and accelerating training process is a critical challenge. Employing our Consistency-FM on training video diffusion models would effectively reduce training costs while keeping high-quality generation.

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

## A  THEORETICAL SUPPORTS AND PROOFS

### A.1  PROOF OF LEMMA 1

*Proof of Lemma 1.* If *Condition 1* is meet, then ODE Eq. (1) associated with $v$ becomes

$$\frac{d\gamma_x(t)}{dt} = v(t, \gamma_x(t)) = v(0, x),$$

and the solution of which is $\gamma_x(t) = x + t * v(0, x)$. Specifically, we have

$$\gamma_x(1) = x + 1 * v(0, x)$$
$$= \gamma_x(t) + (1-t) * v(0, \gamma_x(0)) = \gamma_x(t) + (1-t) * v(t, \gamma_x(t))$$
$$= \gamma_x(s) + (1-s) * v(0, \gamma_x(0)) = \gamma_x(s) + (1-s) * v(t, \gamma_x(s))$$

and thus *Condition 2* is meet.

On the other hand, if *Condition 2* is meet, then we have

$$\gamma_x(t) - \gamma_x(s) = (1-s)v(s, \gamma_x(s)) - (1-t)v(t, \gamma_x(t))$$
$$= \int_s^t v(u, \gamma_x(u))du$$

Divide both hands in the above equation with $t - s$ and let $t$ approaches $s$, we have:

$$v(s, \gamma_x(s)) = \lim_{t \to s} \frac{\int_s^t v(u, \gamma_x(u))du}{t - s},$$
$$= \lim_{t \to s} \frac{(1-s)v(s, \gamma_x(s)) - (1-t)v(t, \gamma_x(t))}{t - s} \tag{13}$$
$$= \lim_{t \to s} v(t, \gamma_x(t)) + (1-s) * \frac{v(s, \gamma_x(s)) - v(t, \gamma_x(t))}{t - s} = v(s, \gamma_x(s)) - (1-s)\frac{dv(s, \gamma_s)}{ds}$$

Comparing the both sides in the above equation, we have $\frac{dv(s, \gamma_s)}{ds} = 0$, and thus *Condition 1* is meet. $\square$

### A.2  PROOF OF LEMMA 2

We provide an another lemma which describes the consistency constraint as partial differential equations and supports the connection bewteen Consistency-FM with FMs.

**Lemma 2.** *Assume $v$ is continuously differentiable and consistent, then $v$ satisfies the following equation:*

$$\partial_t v(t, x) + v \cdot \nabla_x v = 0 \tag{14}$$

*Proof of Lemma 2.* By Lemma 1, if $v$ is consistent then it can be written as:

$$v(t + \Delta t, x_t + \Delta t * v(t, x_t)) = v(t, x_t) \tag{15}$$

Since $v$ is differentiable, we can take derivatives with respect to $t$:

$$\frac{dv}{dt} = \partial_t v + v \cdot \nabla_x v \tag{16}$$

Then by definition, if $v$ is consistent we have

$$\frac{dv}{dt} = \partial_t v + v \cdot \nabla_x v = 0 \tag{17}$$

$\square$

## A.3 PROOF OF THEOREM 1

*Proof of Theorem 1.* By the first mean value theorem, there exist a $t' \in [t, t + \Delta t]$, such that

$$x_{t+\Delta t} - x_t = \int_t^{t+\Delta t} u(s, x_s) du = \Delta t * u(t', x_{t'}) \qquad (18)$$

Then by the definition of $f_\theta(t, x_t) = x_t + (1 - t) * v_\theta(t, x_t)$, we have:

$$
\begin{aligned}
f_\theta(t, x_t) - f_\theta(t + \Delta t, x_{t+\Delta t}) &= x_t + (1 - t) * v_\theta(t, x_t) - (x_{t+\Delta t} + (1 - t - \Delta t) * v_\theta(t + \Delta t, x_{t+\Delta t})) \\
&= x_t - x_{t+\Delta t} + (1 - t) * v_\theta(t, x_t) - (1 - t - \Delta t) * v_\theta(t + \Delta t, x_{t+\Delta t}) \\
&= -\int_t^{t+\Delta t} u(s, x_s) ds + \Delta t * v_\theta(t + \Delta t, x_{t+\Delta t}) + (1 - t) * (v_\theta(t, x_t) - v_\theta(t + \Delta t, x_{t+\Delta t})) \\
&=^1 \Delta t * (v_\theta(t + \Delta t, x_{t+\Delta t}) - u(t', x_{t'})) + (1 - t) * (v_\theta(t, x_t) - v_\theta(t + \Delta t, x_{t+\Delta t})) \\
&=^2 \Delta t * (v_\theta(t + \Delta t, x_{t+\Delta t}) - u(t', x_{t'})) - (1 - t) * \Delta t * (\partial_t v_\theta(t, x_t) - u(t, x_t) \partial_x v_\theta(t, x_t)) + O((\Delta t)^2) \\
&=^3 \Delta t * (v_\theta(t + \Delta t, x_{t+\Delta t}) - u(t, x_t) + O(\Delta t)) - (1 - t) * \Delta t * (\partial_t v_\theta(t, x_t) - u(t, x_t) \partial_x v_\theta(t, x_t)) + O((\Delta t)^2) \\
&= \Delta t * (v_\theta(t + \Delta t, x_{t+\Delta t}) - u(t, x_t) - (1 - t) * (\partial_t v_\theta(t, x_t) - u(t, x_t) \partial_x v_\theta(t, x_t))) + O((\Delta t)^2)
\end{aligned}
\qquad (19)
$$

where in $(1)$ we used the first mean value theorem, in $(2)$ we used first-order Taylor approximation and the boundedness of $u$, $v_\theta$ and their derivatives, in $(3)$ we used first-order Taylor approximation again and the boundedness of derivative of $u$. Then the objective can be written as:

$$
\begin{aligned}
E||f_\theta(t, x_t) - f_\theta(t + \Delta t, x_{t+\Delta t})||_2^2, \\
&= E||\Delta t * (v_\theta(t, x_t) - u(t, x_t) - (1 - t)(\partial_t v_\theta + u \cdot \nabla_x v_\theta)) + O((\Delta t)^2)||_2^2 \\
&= (\Delta t)^2 * E||v_\theta(t, x_t) - u(t, x_t) - (1 - t)(\partial_t v_\theta + u \cdot \nabla_x v_\theta)||_2^2 + o((\Delta t)^2)
\end{aligned}
$$

$\square$

## A.4 PROOF OF THEOREM 2

*Proof of Theorem 2.* As $v_\theta^i(t, x_t)$ is the minimizer of Eq. (8) with respect to segment $i$, it must satisfies the first-order condition:

$$
\begin{aligned}
0 &= \partial_\theta(E||f_\theta^i(t, x_t) - f_{\theta-}^i(t + \Delta t, x_{t+\Delta t})||_2^2 + \alpha||v_\theta^i(t, x_t) - v_{\theta-}^i(t + \Delta t, x_{t+\Delta t})||_2^2) \\
&= E((T - t)(f_\theta^i(t, x_t) - f_{\theta-}^i(t + \Delta t, x_{t+\Delta t})) + \alpha(v_\theta^i(t, x_t) - v_{\theta-}^i(t + \Delta t, x_{t+\Delta t}))) \cdot \partial_\theta v_\theta^i(t, x_t),
\end{aligned}
\qquad (20)
$$

where $f_\theta^i(t, x_t) = x_t + (T - t)v_\theta^i(t, x_t)$ Note that in our assumption, $x_{t+\Delta t}$ is generated by an ODE and thus is a deterministic function of $(t, x_t)$, then the non-trivial solution to 20 satisfies the following equation almost everywhere:

$$0 = (T - t)(f_\theta^i(t, x_t) - f_{\theta-}^i(t + \Delta t, x_{t+\Delta t})) + \alpha(v_\theta^i(t, x_t) - v_{\theta-}^i(t + \Delta t, x_{t+\Delta t})), \qquad (21)$$

As the gradient at $\theta$ is zero, then $\theta = \theta^-$, thus the learned velocity can be derived from 21:

$$v_\theta^i(t, x_t) = \frac{(T - t)(x_{t+\Delta t} - x_t)}{(T - t)^2 + \alpha} + \frac{(T - t - \Delta t)(T - t) + \alpha}{(T - t)^2 + \alpha} v_\theta^i(t + \Delta t, x_{t+\Delta t}) \qquad (22)$$

Furthermore, we have:

$$v_\theta^i(t, x_t)$$

$$= \frac{(T-t)(x_{t+\Delta t} - x_t)}{(T-t)^2 + \alpha} + \frac{(T-t-\Delta t)(T-t) + \alpha}{(T-t)^2 + \alpha} v^*(t + \Delta t, x_{t+\Delta t})$$

$$+ \frac{(T-t-\Delta t)(T-t) + \alpha}{(T-t)^2 + \alpha} v_\theta^i(t + \Delta t, x_{t+\Delta t}) - \frac{(T-t-\Delta t)(T-t) + \alpha}{(T-t)^2 + \alpha} v^*(t + \Delta t, x_{t+\Delta t})$$

$$= \frac{(T-t)((T - (t+\Delta t))v^*(t + \Delta t, x_{t+\Delta t}) + x_{t+\Delta t} - x_t)}{(T-t)^2 + \alpha} + \frac{\alpha v^*(t + \Delta t, x_{t+\Delta t})}{(T-t)^2 + \alpha}$$

$$+ \frac{(T-t-\Delta t)(T-t) + \alpha}{(T-t)^2 + \alpha}(v_\theta^i(t + \Delta t, x_{t+\Delta t}) - v^*(t + \Delta t, x_{t+\Delta t}))$$

$$=^1 \frac{(T-t)(x_T - x_t)}{(T-t)^2 + \alpha} + \frac{\alpha v^*(t + \Delta t, x_{t+\Delta t})}{(T-t)^2 + \alpha}$$

$$+ \frac{(T-t-\Delta t)(T-t) + \alpha}{(T-t)^2 + \alpha}(v_\theta^i(t + \Delta t, x_{t+\Delta t}) - v^*(t + \Delta t, x_{t+\Delta t}))$$

$$=^2 \frac{(T-t)^2 v^*(t, x_t)}{(T-t)^2 + \alpha} + \frac{\alpha v^*(t + \Delta t, x_{t+\Delta t})}{(T-t)^2 + \alpha}$$

$$+ \frac{(T-t-\Delta t)(T-t) + \alpha}{(T-t)^2 + \alpha}(v_\theta^i(t + \Delta t, x_{t+\Delta t}) - v^*(t + \Delta t, x_{t+\Delta t}))$$

$$= v^*(t, x_t) + \frac{\alpha(v^*(t + \Delta t, x_{t+\Delta t}) - v^*(t, x_t))}{(T-t)^2 + \alpha}$$

$$+ \frac{(T-t-\Delta t)(T-t) + \alpha}{(T-t)^2 + \alpha}(v_\theta^i(t + \Delta t, x_{t+\Delta t}) - v^*(t + \Delta t, x_{t+\Delta t}))$$

$$\tag{23}$$

where in (1) and (2) we have use the assumption of oracle $v^*$ that $x_T = x_t + (T-t)v^*(t, x_t)$ □

## A.5 PROOF OF COROLLARY 2.1

*Proof for Corollary 2.1.* Note that $x_T = x_T + 0 * v_\theta^i(T, x_T)$, and thus we can set arbitrary value for $v_\theta^i(T, x_T)$ without affecting the model. Specifically, we set $v_\theta^i(T, x_T) = v^*(T, x_T)$. Then by Theorem 2, the error at $T - \Delta t$ is:

$$v_\theta^i(T - \Delta t, x_{T-\Delta t}) - v^*(T - \Delta t, x_{T-\Delta t})$$

$$= \frac{\alpha}{(\Delta t)^2 + \alpha}(v^*(T, x_T) - v^*(T - \Delta t, x_{T-\Delta t}))$$

$$+ \frac{(\Delta t - \Delta t)(T - t) + \alpha}{(\Delta t)^2 + \alpha}(v_\theta^i(T, x_T) - v^*(T, x_T)) \tag{24}$$

$$= \frac{\alpha}{(\Delta t)^2 + \alpha}(v^*(T, x_T) - v^*(T - \Delta t, x_{T-\Delta t}))$$

Furthermore, as $u$ is consistent within $[S, T]$, by Lemma 1 we have :

$$x_T = x_t + (T-t) * u(t, x_t),$$

$$\Rightarrow v^*(t, x_t) = u(t, x_t) \equiv u(T, x_T),$$

$$\Rightarrow v^*(t, x_t) = v^*(t + \Delta t, x_{t+\Delta t}), \forall t \in [S, T - \Delta t].$$

And thus $v_\theta^i(T - \Delta t, x_{T-\Delta t}) - v^*(T - \Delta t, x_{T-\Delta t}) = 0$.

By Theorem 2, we can deduce by induction that

$$v_\theta^i(t + \Delta t, x_{t+\Delta t}) = v^*(t + \Delta t, x_{t+\Delta t}) \quad \& \quad v^*(t, x_t) = v^*(t + \Delta t, x_{t+\Delta t})$$

$$\Rightarrow v_\theta^i(t, x_t) = v^*(t, x_t) = u(t, x_t), \forall t \in [S, T - \Delta t].$$

□

## A.6 New Theorem based on Consistency model Thm. 2

Here we provide an intuition why training Consistency-FM from scratch works. In the following result, we show that training Consistency-FM from scratch using conditional trajectory $x_t, x_{t+\Delta t}$ is equivalent to training with a perfectly pre-trained FM trajectory, as $\Delta t \approx 0$. As a result, Consistency-FM can learn from the underlying probability path without accessing the ground truth velocity. Without loss of generality, we only consider one-segment setting. Let

$$\mathcal{L}_\theta = E||f_\theta(t, x_t) - f_{\theta^-}(t + \Delta t, x_{t+\Delta t})||_2^2,$$

which is the first part of Consistency-FM's loss function at time $t$, and $x_t = tx_1 + (1-t)x_0$ and $x_{t+\Delta t} = (t + \Delta t)x_1 + (1 - t - \Delta t)x_0$. And then we define the loss function that utilize pre-trained FM:

$$\mathcal{L}_{FM} = E||f_\theta(t, x_t) - f_{\theta^-}(t + \Delta t, x_{t+\Delta t}^{FM})||_2^2,$$

where $x_{t+\Delta t}^{FM}$ is generated by an perfectly pre-trained FM model: $u_\phi(x_t, t) = E(x_1 - x_0|x_t)$, and thus $x_{t+\Delta t}^{FM} = x_t + \int_t^{t+\Delta t} u_\phi(x_s, s)ds$. Then we have:

**Theorem 3.** *Assume $f_\theta$, $f_{\theta^-}$ and $u_\phi(x_t, t)$ are bounded, $f_{\theta^-}$ is twice-continuous differentiable and has bounded second derivatives, then*

$$\mathcal{L}_{FM} = \mathcal{L}_\theta + o(\Delta t) \tag{25}$$

*Proof.* We expand $\mathcal{L}_{FM}$ using first-order Taylor expansion with respect to $\Delta t$:

$$\mathcal{L}_{FM} = E||f_\theta(t, x_t) - f_{\theta^-}(t + \Delta t, x_{t+\Delta t}^{FM})||_2^2$$
$$= E(f_\theta(t, x_t) - f_{\theta^-}(t, x_t)^T(f_\theta(t, x_t) - \partial_t f_{\theta^-}(t, x_t)\Delta t + \partial_x f_{\theta^-}(t, x_t)u_\phi(x_t, t)\Delta t + o(\Delta t))$$
$$=^1 E(f_\theta(t, x_t) - f_{\theta^-}(t, x_t)^T(f_\theta(t, x_t) - \partial_t f_{\theta^-}(t, x_t)\Delta t + \partial_x f_{\theta^-}(t, x_t)u_\phi(x_t, t)\Delta t) + o(\Delta t)$$
$$= E(f_\theta(t, x_t) - f_{\theta^-}(t, x_t)^T(f_\theta(t, x_t) - \partial_t f_{\theta^-}(t, x_t)\Delta t + \partial_x f_{\theta^-}(t, x_t)E(x_1 - x_0|x_t)\Delta t) + o(\Delta t)$$
$$=^2 E(f_\theta(t, x_t) - f_{\theta^-}(t, x_t)^T(f_\theta(t, x_t) - \partial_t f_{\theta^-}(t, x_t)\Delta t + \partial_x f_{\theta^-}(t, x_t)(x_1 - x_0)\Delta t) + o(\Delta t)$$
$$=^3 E||f_\theta(t, x_t) - f_{\theta^-}(t + \Delta t, x_t + (x_1 - x_0)\Delta t)||_2^2 + o(\Delta t)$$
$$= E||f_\theta(t, x_t) - f_{\theta^-}(t + \Delta t, x_{t+\Delta t})||_2^2 + o(\Delta t)$$
$$= \mathcal{L}_\theta + o(\Delta t),$$
$$\tag{26}$$

where in (1) we used the boundedness, in (2) we used the law of total expectation, and in (3) we used the first-order Taylor expansion again. $\square$

## B More Implementation Details and Comparison Results

Table 5: Experimental details for training Consistency-FM.

| Training Details | CIFAR-10 | AFHQ-Cat | CelebA-HQ | ImageNet |
|---|---|---|---|---|
| Training iterations | 180k | 250k | 250k | 30K |
| Batch size | 512 | 64 | 64 | 128 |
| Optimizer | Adam | Adam | Adam | Adam |
| Learning rate | 2e-4 | 2e-4 | 2e-4 | 8e-6 |
| $\Delta t$ | 0.001 | 0.001 | 0.001 | 0.001 |
| EMA decay rate | 0.999999 | 0.999 | 0.999 | 0.999 |
| ODE solver | Euler | Euler | Euler | Euler |

Table 6: Comparison of FID on MS COCO 2014 following the evaluation setup in (Liu et al., 2023).

| Cat. | Res. | Method | Inference Time | # Param. | FID-30k |
|---|---|---|---|---|---|
| AR | 256 | Parti-750M (Yu et al., 2022) | - | 750M | 10.71 |
| AR | 256 | Parti-3B (Yu et al., 2022) | 6.4s | 3B | 8.10 |
| AR | 256 | Parti-20B (Yu et al., 2022) | - | 20B | 7.23 |
| AR | 256 | Make-A-Scene (Gafni et al., 2022) | 25.0s | - | 11.84 |
| Diff | 256 | GLIDE (Nichol et al., 2022) | 15.0s | 5B | 12.24 |
| Diff | 256 | LDM (Rombach et al., 2022) | 3.7s | 0.27B | 12.63 |
| Diff | 256 | DALL-E 2 (Ramesh et al., 2022) | - | 5.5B | 10.39 |
| Diff | 256 | Imagen (Saharia et al., 2022) | 9.1s | 3B | 7.27 |
| Diff | 256 | eDiff-I (Balaji et al., 2022) | 32.0s | 9B | 6.95 |
| - | 512 | Muse-3B (Chang et al., 2023) | 1.3s | 0.5B | 7.88 |
| GAN | 512 | StyleGAN-T (Sauer et al., 2023) | 0.10s | 1B | 13.90 |
| GAN | 512 | GigaGAN (Kang et al., 2023) | 0.13s | 1B | 9.09 |
| Diff | 512 | SD (Rombach et al., 2022) | 2.9s | 0.9B | 9.62 |
| Diff | 512 | DMD (Yin et al., 2024) | 0.09s | 0.9B | 11.49 |
| FM | 512 | Rectified-Flow (Liu et al., 2022) | 0.09s | 0.9B | 13.67 |
| FM | 512 | InstaFlow (Liu et al., 2023) | 0.09s | 0.9B | 13.10 |
| FM | 512 | PeRFlow (Yan et al., 2024) | 0.09s | 0.9B | 18.59 |
| FM | 512 | **Consistency-FM** | 0.09s | 0.9B | **11.02** |

Table 7: We measure the generation diversity as Sadat et al. (2024) for more comprehensive evaluations. Specifically, given a set of input conditions, we first compute the pairwise cosine similarity matrix K among generated images with the same condition, using SSCD (Pizzi et al., 2022) as the pretrained feature extractor. The results are then aggregated for different conditions using two methods: the Mean Similarity Score (MSS), which is a simple average over the similarity matrix K , and the Vendi Score (Friedman & Dieng, 2022), which is based on the Von Neumann entropy of K.

| Diversity Metric | MSS ↓ | Vendi Score ↑ |
|---|---|---|
| Rectified Flow (Liu et al., 2022) | 0.35 | 3.21 |
| Consistency Model (Song et al., 2023) | 0.24 | 4.87 |
| **Consistency-FM (Ours)** | **0.19** | **5.33** |

