# OpenReview forum: "Consistency Flow Matching: Defining Straight Flows with Velocity Consistency"
_ICLR.cc/2025/Conference — Submitted to ICLR 2025_

### Official Review · Reviewer_Yzev · 2024-10-31

**Soundness:** 3
**Presentation:** 3
**Contribution:** 3
**Rating:** 6
**Confidence:** 4

**Summary:**

This paper presents Consistency Flow Matching (Consistency-FM) for defining probability paths to transform between noise and data samples. Consistency-FM enforces self-consistency in the velocity field, defining straight flows from different times to the same endpoint with velocity constraints. A multi-segment training approach is proposed to enhance expressiveness, optimizing the trade-off between sampling quality and speed. Experiments demonstrate that Consistency-FM improves training efficiency while delivering better generation quality.

**Strengths:**

The design approach based on velocity consistency introduces novel methods for optimizing, distilling, and sampling diffusion-based models in detail.

The idea of multi-segment optimization is very novel and effective. Additionally, the theoretical derivation of the training objectives is provided in detail. Experimental results, including FID and CLIP scores, demonstrate the superiority of the proposed methods.

**Weaknesses:**

The comparison of visualized samples does not show the advantages of the proposed method clearly. Visual quality is judged by humans based on a few samples, and diversity is also a critical metric.

Comparisons of training time to model convergence are important for a comprehensive evaluation but are missing in this paper. Furthermore, I suggest the authors provide a unified metric to measure both quality and speed.

**Questions:**

In Theorem 1, the authors consider no exponential moving average, and the ground truth velocity field is assumed to be known. I recommend discussing the impact of these assumptions on practical applications.

The role of the second term in Equation 6 needs further explanation. The first term's role in striking a balance between exact velocity estimation and adhering to consistent velocity constraints is well explained in Remark 1. A similar explanation is needed for the second term.

---

> ### Author Response · Authors · 2024-11-20
> **Response to Reviewer Yzev - Part 1**
>
> *Thank you for your detailed feedback and for taking the time to review our paper. We appreciate your valuable insights and suggestions. Please see below for our responses to your comments and we have updated the manuscript to reflect these improvements and clarifications.*
>
> **Q1: Visual quality is judged by humans based on a few samples, and diversity is also a critical metric.**
>
> A1: In addition to our visualized comparisons, we also provide quantitative results on text-to-image generation in the Table 5 and Table 6 of our paper, and we can find that our Consistency-FM significantly outperforms previous methods, demonstrating our superiority. To follow your suggestions, we here measure the generation diversity as [1] for more comprehensive evaluations. Specifically, given a set of input conditions, we first compute the pairwise cosine similarity matrix K among generated images with the same condition, using SSCD [2] as the pretrained feature extractor. The results are then aggregated for different conditions using two methods: the Mean Similarity Score (MSS), which is a simple average over the similarity matrix K , and the Vendi Score [3], which is based on the Von Neumann entropy of K. We list the results in the table below. We also **update this results in the Table 7 of our revised manuscript**, thanks for your suggestions.
>
> |Diversity Metric|MSS ↓| Vendi Score ↑|
> | ----| ------| ---|
> |Rectified Flow| 0.35| 3.21|
> |Consistency Model|0.24| 4.87|
> |**Consistency-FM**|**0.19** |**5.33**|
>
> From the results, we can find that our Consistency-FM is consistently better than previous methods regarding the diversity metric, demonstrating the expressiveness of our velocity consistency and multi-segment design.
>
> [1] Sadat, Seyedmorteza, et al. "CADS: Unleashing the Diversity of Diffusion Models through Condition-Annealed Sampling." ICLR 2024.
>
> [2] Pizzi E, Roy S D, Ravindra S N, et al." A self-supervised descriptor for image copy detection." CVPR 2022.
>
> [3] Friedman D, Dieng A B. "The Vendi Score: A Diversity Evaluation Metric for Machine Learning." TMLR 2022.
>
> **Q2: Comparisons of training time to model convergence are important for a comprehensive evaluation but are missing in this paper. Furthermore, I suggest the authors provide a unified metric to measure both quality and speed.**
>
> A2: We appreciate your insightful suggestion. Below, we provide a comparison of training times to model convergence on the CIFAR-10 dataset for Consistency-FM, Consistency Model, and RectifiedFlow:
>
> | Method                        | Consistency Model | RectifiedFlow | Ours |
> | ---------- | ------- | ------------- | ---- |
> | Training time (A100 GPU days) | 38             | 13         | 8 |
>
> From the comparison, we can find that our Consistency-FM is indeed more efficient for training due to our proposed velocity consistency.
>
> Regarding the unified metric to measure both quality and speed, we acknowledge the importance of such a metric for comprehensive evaluation. Currently, there is no widely accepted unified metric, and to ensure fair comparison, we have not used one in our study. However, we recognize the value of this metric and will consider designing a unified metric to measure both quality and speed in future work. Thank you for this valuable suggestion.

---

> ### Author Response · Authors · 2024-11-20
> **Response to Reviewer Yzev - Part 2**
>
> **Q3: In Theorem 1, the authors consider no exponential moving average, and the ground truth velocity field is assumed to be known. I recommend discussing the impact of these assumptions on practical applications.**
>
> A3: Thank you for raising this important point. We address the impact of these assumptions on practical applications as follows:
>
> In practical experiments, we found that training with  $\theta^-$ replaced with $stopgrad(\theta)$ leads to a slight improvement in FID. Therefore, considering no exponential moving average does not significantly affect the validity of Theorem 1.
>
> In the updated Appendix A.6, highlighted in blue, we provide Theorem 3, which shows that training Consistency-FM using the conditional trajectory $x_t, x_{t+\Delta t}$ is equivalent to training with a perfectly pre-trained FM trajectory that estimates the ground truth velocity field. The equation below is from the updated Theorem 3:
>
> $$
> E|| f\_\theta(t,x\_{t}) - f\_{\theta^-}(t+\Delta t,x\_{t+\Delta t}^{FM})||\_2^2 = E|| f\_\theta(t,x\_{t}) - f\_{\theta^-}(t+\Delta t,x\_{t+\Delta t})||\_2^2 + o(\Delta t)
> $$
>
> where $x_{t+\Delta t}^{FM}$ is generated by a perfectly pre-trained FM model. As a result, Consistency-FM can learn from the underlying probability path without accessing the ground truth velocity. This demonstrates that the assumptions made in Theorem 1 do not hinder the practical applicability of our method.
>
> **Q4: The role of the second term in Equation 6 needs further explanation.**
>
> A4: It should be noted that our velocity consistency involves two terms in our loss function for differnt reasons. The first term is performed in the sample space with our defined straight flows (for improving both performance and effciency), and the second term directly constrains on velocity consistency (for better trade-off between performance and effciency). As shown in the table below, we conduct additional ablation experiments on the CIFAR-10 dataset.
>
> | Ablation study | Consistency model (traditional sample-space consistency)| Ours ($\alpha$=0)    | Ours ($\alpha$=1e-5) | Ours ($\alpha$=1e-4) |
> | ------------------------------------ |:----: | :----: | :----: | :----: |
> | FID                                  |5.83| 5.29 | 5.34 | 5.43 |
> | Training steps ($\times 10^3$)     | 800|225  | 175  | 145  |
>
> From the results, we can observe that when $\alpha=0$ (using only $\left|f_\theta - f_{\theta^{-}}\right|$, Condition 2), both FID and efficiency can be improved, better than traditional sample-space consistency. With an increase in the weight $\alpha$ for velocity difference $(\left\|v_\theta(t, x_t) - u(t, x_t)\right\|_2^2$, Condition 1), we observed a slight degradation in FID but with the significant enhanced training efficiency.
> This demonstrates that the sample space defined with our velocity consistency (first term) indeed improves both sampling quality and training efficiency while the direct constraint on velocity space (second term) can result in a better trade-off between model performance and convergence. Therefore, both loss are critical for our Consistecy-FM.

---

> > ### Comment · Reviewer_Yzev · 2024-11-27
> >
> > Thank you for the point-by-point response, which addressed my concerns. I will maintain my original positive rating and vote to accept this paper.

---

> > > ### Author Response · Authors · 2024-11-27
> > > **Thanks for your support**
> > >
> > > Dear Reviewer Yzev，
> > >
> > > Thanks for maintaining original positive rating! We greatly appreciate your recognition of our work and the valuable feedback you provided
> > >
> > > Warm Regards,
> > >
> > > The Authors

---

### Official Review · Reviewer_4ccL · 2024-11-02

**Soundness:** 3
**Presentation:** 2
**Contribution:** 3
**Rating:** 6
**Confidence:** 4

**Summary:**

The paper proposes Consistency-FM, a method to train generative models in the flow matching framework with an explicit regularization that imposes consistent learned vector fields and hence enables faster inference. The method can be both trained from scratch and distilled from a pre-trained generator. Additionally, self-consistency can be enforced on disjoint trajectory segments to improve the expressiveness of the trained generator. Through experiments the method is empirically shown to converge faster than prior work and produce better quality samples with fewer denoising steps.

**Strengths:**

The proposed approach is novel and interesting. It produces great results even without distillation, which is a significant step forward in making iterative denoising methods efficient at inference. Theoretical analysis of the proposed loss is provided.

**Weaknesses:**

The presentation of the paper can be improved. Certain points are rather unclear, as some details are missing from the paper (see Questions section). The paper does not provide any ablation studies or highlight the significance of certain design choices in any form except for motivation (e.g. lines 283-286 for multisegment consistency-fm). This makes it difficult to understand what exactly leads to the results reported in the paper. Nevertheless, I am slightly leaning towards acceptance of the paper, as the strengths outweigh the above weaknesses. However, I expect the authors to address my concerns below in the rebuttal and edit the manuscript accordingly by providing more details. I will update my final rating based on the rebuttal and the other reviews.

**Questions:**

Here are some questions and concerns:

1) According to Lemma 1 both conditions are equivalent. While lines 246-247 provide an explanation of why Condition 1 cannot be used alone, the intuition why Condition 2 cannot be used alone is missing. Why is there a need to use both in the loss? Some ablation studies on this could be helpful.
2) In Lemma 1 both conditions are imposed with arbitrary $t$ and $s$. Why in the loss only neighboring points are considered? How does the performance depend on the choice of $\Delta t$?
3) Based on the proof in the Appendix A.3, in Theorem 1, $x_t$ and $x_{t + \Delta t}$ are points on a generative trajectory induced by the ground truth velocity field $u_t$. However, during training, especially in the non-distillation case, this ground truth probability path is unavailable and $x_t$ and $x_{t + \Delta t}$ are sampled from the conditional probability paths. Because of this difference, it seems that Theorem 1 cannot provide intuition on why training from scratch works. Could the authors clarify this?
4) Based on Theorem 1, there is a trade-off between consistency and matching the ground truth vector field. However, this trade-off is not observed in the experiments, where Consistency-FM is always strictly better than the prior work. Could the authors comment on this?
5) Multi-segment Consistency-FM is an interesting extension of the proposed method. Unfortunately, its effect on the results is not demonstrated in the paper. Some ablation studies, e.g. on the number of segments and their sizes, would definitely improve the presentation quality.
6) From the paper and the implementation details in section 4 it is not clear, which configuration of the method was used for the experiments. E.g. what is the value of $\Delta t$? Was multi-segment consistency-fm used? Is the number of the segments equal to the NFE in Tables?
7) Assuming that the models are trained with the multi-segment approach, can a model trained with $K$ segments be used with a different number of NFE at inference?
8) How exactly is the GAN loss integrated in the training?
9) Some points in the proofs could be expanded. E.g. step 3 in line 829.

Some typos:
1) L093 - sample with one NFE, ~~but~~ they fail
2) L097 - While CMs ~~is~~ are able
3) L107 - Further ~~text-ti-image~~ text-to-image experiments
4) L172 - However, these ~~method~~ methods
5) L298 - First, we ~~analysis~~ analyse
6) L299 - provide ~~a~~ an explicit formula

---

> ### Author Response · Authors · 2024-11-20
> **Response to Reviewer 4ccL - Part 1**
>
> *Thank you for your detailed feedback and for taking the time to review our paper. We appreciate your valuable insights and suggestions. Please see below for our responses to your comments and we have updated the manuscript to reflect these improvements and clarifications.*
>
> **Q1: Why is there a need to use Condition 1 in the loss? Some ablation studies on this could be helpful.**
>
> A1: It should be noted that our velocity consistency involves two terms in our loss function for different reasons. The first term is performed in the sample space with our defined straight flows (for improving both performance and efficiency), and the second term directly constrains on velocity consistency (for better trade-off between performance and efficiency). As shown in the table below, we conduct additional ablation experiments on the CIFAR-10 dataset.
>
> | Ablation study | Consistency model (traditional sample-space consistency)| Ours ($\alpha$=0)    | Ours ($\alpha$=1e-5) | Ours ($\alpha$=1e-4) |
> | ------------------------------------ |:----: | :----: | :----: | :----: |
> | FID                                  |5.83| 5.29 | 5.34 | 5.43 |
> | Training steps ($\times 10^3$)     | 800|225  | 175  | 145  |
>
> From the results, we can observe that when $\alpha=0$ (using only $\left|f_\theta - f_{\theta^{-}}\right|$, Condition 2), both FID and efficiency can be improved, better than traditional sample-space consistency. With an increase in the weight $\alpha$ for velocity difference ($\left\|v_\theta(t, x_t) - u(t, x_t)\right\|_2^2$, Condition 1), we observed a slight degradation in FID but with the significant enhanced training efficiency.
> This demonstrates that the sample space defined with our velocity consistency (first term) indeed improves both sampling quality and training efficiency while the direct constraint on velocity space (second term) can result in a better trade-off between model performance and convergence. Therefore, both loss are critical for our Consistency-FM.
>
>
> **Q2: In Lemma 1 both conditions are imposed with arbitrary t and s. Why in the loss only neighboring points are considered? How does the performance depend on the choice of Δt?**
>
> A2: When each point and its neighboring point satisfy the conditions in Lemma 1, the conditions are also satisfied for arbitrary points at t and s. Arbitrarily choosing t and s implies that during training, Δt = s - t will be arbitrary. However, our preliminary experiments indicated that training converged more unstably when Δt was either excessively large or too small. Since we multiply the timestep by 999 before converting it into a positional time embedding, following the implementation of Rectified Flow [1], and based on our early experimental results, we use Δt = 1e-3 in the final experiments. Below is a table showing how performance depends on the choice of Δt:
>
> | Δt             | 1e-4  | 1e-3 | 1e-2  | 1e-1  |
> | -------------- | ----- | ---- | ----- | ----- |
> | FID            | 94.34 | 5.34 | 7.42  | 22.65 |
> | Training steps ($\times 10^3$) | 87.5  | 175  | 137.5 | 112.5 |
>
>
>
> [1] Liu, Xingchao, and Chengyue Gong. "Flow Straight and Fast: Learning to Generate and Transfer Data with Rectified Flow." International Conference on Learning Representations, 2023.
>
> **Q3: It seems that Theorem 1 cannot provide intuition on why training from scratch works. Could the authors clarify this?**
>
> A3: Thank you for your insightful comment. We provide an intuition on why training Consistency-FM from scratch works **in the updated Appendix A.6**, highlighted in blue. By proving Theorem 3, we show that training Consistency-FM from scratch using the conditional trajectory $x_t, x_{t+\Delta t}$ is equivalent to training with a perfectly pre-trained FM trajectory, as $\Delta t \approx 0$. The equation below is from the updated Theorem 3. Here, $x_{t+\Delta t}^{FM}$ is generated by a perfectly pre-trained FM model.
> $$
> E|| f\_\theta(t,x\_{t}) - f\_{\theta^-}(t+\Delta t,x\_{t+\Delta t}^{FM})||\_2^2 = E|| f\_\theta(t,x\_{t}) - f\_{\theta^-}(t+\Delta t,x\_{t+\Delta t})||\_2^2 + o(\Delta t)
> $$
> As a result, Consistency-FM can learn from the underlying probability path without accessing the ground truth velocity.
>
>
>
> **Q4: Based on Theorem 1, there is a trade-off between consistency and matching the ground truth vector field. However, this trade-off is not observed in the experiments, where Consistency-FM is always strictly better than the prior work. Could the authors comment on this?**
>
> A4: As demonstrated in the table provided in above Answer 1, Consistency-FM achieves a balance between performance and training efficiency by navigating the trade-off between consistency and matching the ground truth vector field. This equilibrium enables Consistency-FM to effectively optimize both model performance and training efficiency.

---

> ### Author Response · Authors · 2024-11-20
> **Response to Reviewer 4ccL - Part 2**
>
> **Q5: Some ablation studies for the multi-segment Consistency-FM, e.g. on the number of segments and their sizes, would definitely improve the presentation quality.**
>
> A5: Thank you for the suggestion. As shown in the table below, sample quality improves with an increase in the number of segments for the multi-segment Consistency-FM model on the CIFAR-10 dataset. Here the number of evaluations (NFE) is equal to the number of segments for the multi-segment Consistency-FM.
>
> | number of segments | 2    | 4    | 6    |
> | ------------------ | ---- | ---- | ---- |
> | FID                | 5.34 | 3.97 | 3.04 |
>
> This demonstrates that increasing the number of segments enhances the quality of generated samples, validating the effectiveness of the multi-segment approach in improving model performance.
>
>
>
> **Q6: From the paper and the implementation details in section 4 it is not clear, which configuration of the method was used for the experiments. E.g. what is the value of $\Delta t$​ ? Was multi-segment consistency-fm used? Is the number of the segments equal to the NFE in Tables?**
>
> A6: We apologize for the oversight in the implementation details. We set $\Delta t = 0.001$ in all experiments. Multi-segment Consistency-FM is applied only for experiments where the number of function evaluations (NFE) is greater than 1. The NFE is equivalent to the number of segments for the multi-segment Consistency-FM. We have added these details to the manuscript for clarity, highlighted in blue. Thank you for bringing this to our attention.
>
>
> **Q7: Assuming that the models are trained with the multi-segment approach, can a model trained with $K$​​ segments be used with a different number of NFE at inference?**
>
> A7: Yes, a model trained with $K$ segments can alternate denoising and noise injection steps similar to Consistency Models in some segments and achieve a different number of evaluations (NFE) at inference. For example, in the i-th segment, we use the following formula to denoise:
> $$
> x_{i / K}=x_{(i-1) / K}+ \frac{1}{K} * v_\theta^i\left(\frac{(i-1)}{K}, x_{(i-1) / K}\right)
> $$
> Then we inject noise and denoise again using the formula, but replacing $x_{(i-1) / K}$ with $x'\_{(i-1) / K}$.
>
> **Q8: How exactly is the GAN loss integrated in the training?**
>
> A8: We integrate the GAN loss in training our Consistency-FM similarly to CTM [1], which is not our main contribution and thus we did not discuss it in our paper. More specifically, we additionally train a discriminator to distinguish between our generated (fake) image and real image, and our generator (sampler) tries to cheat the discriminator by learning the real data distribution as much as possible. This adversarial loss shows that a combination of reconstruction and adversarial losses is beneficial for generation quality.
>
>
>
> [1] Kim, Dongjun, et al. "Consistency trajectory models: Learning probability flow ode trajectory of diffusion." *arXiv preprint arXiv:2310.02279* (2023).
>
>
>
> **Q9: Some points in the proofs could be expanded. E.g. step 3 in Equation 19.**
>
> A9: We have extended the proof of Theorem 1 **in the updated Appendix A.3**. The revised sections are highlighted in blue:
> $$
> \begin{aligned}
>  f_\theta(t,x_t) - f_\theta(t+\Delta t , x_{t+\Delta t}) &= x_t +(1-t)*v_\theta(t,x_t) - (x_{t+\Delta t} +  (1-t-\Delta t)*v_\theta(t+\Delta t, x_{t+\Delta t}) ) \\\\
> &= x_t - x_{t+\Delta t} + (1-t) * v_\theta(t,x_t)  - (1-t-\Delta t) * v_\theta (t+\Delta t , x_{t+\Delta t}) \\\\
> &= -\int_{t}^{t+\Delta t} u(s,x_s)ds + \Delta t * v_\theta (t+\Delta t, x_{t+\Delta t}) + (1-t) * (v_\theta (t,x_t)-v_\theta (t+\Delta t, x_{t+\Delta t}))\\\\
> &= \Delta t * (v_\theta (t+\Delta t, x_{t+\Delta t}) - u(t^{'},x_{t^{'}})) + (1-t) * (v_\theta (t,x_t)-v_\theta (t+\Delta t, x_{t+\Delta t}))  \\\\
> &=\Delta t * (v_\theta (t+\Delta t, x_{t+\Delta t}) - u(t^{'},x_{t^{'}})) - (1-t) * \Delta t * (\partial_t v_\theta (t,x_t)-u(t,x_t)\partial_x v_\theta (t, x_{t})) + O((\Delta t)^{2}) \\\\
> &=\Delta t * (v_\theta (t+\Delta t, x_{t+\Delta t}) - u(t,x_{t})+O(\Delta t)) - (1-t) * \Delta t * (\partial_t v_\theta (t,x_t)-u(t,x_t)\partial_x v_\theta (t, x_{t})) + O((\Delta t)^{2}) \\\\
> &= \Delta t * (v_\theta (t+\Delta t, x_{t+\Delta t}) - u(t,x_{t}) - (1-t) * (\partial_t v_\theta (t,x_t)-u(t,x_t)\partial_x v_\theta (t, x_{t})) )+ O((\Delta t)^{2})
> \end{aligned}
> $$
>
>
> Thank you for pointing out the typos in our paper. We have corrected them accordingly. We appreciate your thorough review and valuable feedback.

---

> > ### Comment · Reviewer_4ccL · 2024-11-26
> >
> > Dear Authors,
> >
> > Thank you for your time and valuable additional clarifications in the rebuttal. Most of my concerns were addressed in the rebuttal. I will keep my original positive rating.
> >
> > Best regards, Reviewer

---

> > > ### Author Response · Authors · 2024-11-26
> > > **Thank you for your support**
> > >
> > > Dear Reviewer 4ccL，
> > >
> > > Thanks for keeping original positive rating! We sincerely appreciate your valuable comments and your precious time in reviewing our paper!
> > >
> > > Warm Regards,
> > >
> > > The Authors

---

### Official Review · Reviewer_AJPE · 2024-11-04

**Soundness:** 2
**Presentation:** 3
**Contribution:** 2
**Rating:** 5
**Confidence:** 4

**Summary:**

The paper proposes integrating a consistency model with a flow matching model by applying an additional velocity field consistency loss alongside the standard sample-space consistency loss. Additionally, it introduces a multi-segment approach to balance sampling quality with efficiency. Experimental results suggest that this method outperforms selected baselines across several benchmarks, demonstrating improved performance.

**Strengths:**

- The paper is clearly written and easy to follow.
- The approach of enforcing consistency in the velocity field is novel.
- The proposed multi-segment Consistency-FM model is a well-considered design intended to balance sampling quality with efficiency.

**Weaknesses:**

- The paper claims that the proposed Consistency-FM model enforces consistency in the velocity field space, unlike the latent consistency model, which applies it directly in the sample space. However, the final consistency loss in Consistency-FM still minimizes the difference in sample space, specifically as $|f_{\theta} - f_{\theta^-}|$. Furthermore, examination of the source code reveals that the weight parameter $\alpha$ for the velocity difference is set to a minimal value (1e-5), calling into question the strength of this claim. These observations suggest that the difference from the standard consistency model is marginal, which may limit the novelty of the approach.
- While flow matching models are generally reported to outperform diffusion models [1], the paper shows only a minor improvement in FID score in Table 1 when compared to the consistency model. It is therefore unclear whether this performance gain arises from replacing the diffusion model with the flow matching model. To clarify this, a baseline using a standard consistency loss (sample space only) in combination with the flow matching model is needed.
- The paper lacks a comprehensive quantitative comparison against prior work. For instance, the CIFAR-10 experiments omit the distillation sampling variant, and the ImageNet 64x64 experiments do not include a direct consistency training variant. Additionally, the ImageNet 64x64 experiments lack a 2 NFE configuration for Consistency-FM. Both the CIFAR-10 and ImageNet experiments omit several key baselines [2][3][4].
- Although the paper introduces a multi-segment Consistency-FM to balance between sample quality and sampling efficiency, it lacks a thorough analysis of how varying NFE steps impact sample quality.

[1] Ma, N., Goldstein, M., Albergo, M.S., Boffi, N.M., Vanden-Eijnden, E. and Xie, S., 2024. Sit: Exploring flow and diffusion-based generative models with scalable interpolant transformers. *arXiv preprint arXiv:2401.08740*.

[2] Berthelot, D., Autef, A., Lin, J., Yap, D.A., Zhai, S., Hu, S., Zheng, D., Talbott, W. and Gu, E., 2023. Tract: Denoising diffusion models with transitive closure time-distillation. *arXiv preprint arXiv:2303.04248*.

[3] Song, Y. and Dhariwal, P., 2023. Improved techniques for training consistency models. *arXiv preprint arXiv:2310.14189*.

[4] Heek, J., Hoogeboom, E. and Salimans, T., 2024. Multistep consistency models. *arXiv preprint arXiv:2403.06807*.

**Questions:**

- Can the authors provide empirical or theoretical evidence illustrating how velocity-space consistency improves model performance compared to traditional sample-space consistency? Additionally, clarifying the impact of velocity-based consistency on the model’s stability or efficiency could strengthen this argument.
- The source code reveals that the weight $\alpha$ for velocity difference is set to 1e-5, a very small value. What was the reasoning behind selecting this weight, and how does it impact the model’s performance?
- The reported FID score improvements over diffusion models appear marginal. Can you provide insights into whether these improvements are statistically significant, or are due to the flow matching model itself?
- Include aforementioned key benchmarks to facilitate a more comprehensive comparison.
- Include an analysis of how sample quality changes with different NFE steps, especially for the multi-segment Consistency-FM model.

---

> ### Author Response · Authors · 2024-11-20
> **Response to Reviewer AJPE - Part 1**
>
> *We sincerely thank you for your time and efforts in reviewing our paper, and your valuable feedback. Please see below for our responses to your comments and we have updated the manuscript to reflect these improvements and clarifications.*
>
> **Q1: Clarify the difference between Consistency-FM and the standard consistency model regarding the final consistency loss. And what is the reasoning behind the weight α for velocity difference is set to 1e-5, and how does it impact the model’s performance?**
>
> A1: It is noted that our Consistency-FM defines straight flows with velocity consistency that innovatively bridges consistency models and flow matching models in a simple yet effective manner:
>
> **Novelty and effectiveness of our first term**: The term $\left|f_\theta - f_{\theta^{-}}\right|$ is derived from Lemma 1, which for the first time defines straight flows with velocity consistency. In Equation (5), these two conditions are equivalent: Condition 1 explicitly enforces velocity consistency, while Condition 2 leads to $\left|f_\theta - f_{\theta^{-}}\right|$. This form is simpler but more effective than the standard consistency model, which has sufficiently demonstrated in Table 1 and Table 2. Additionally, Theorem 1 proves that $\left|f_\theta - f_{\theta^{-}}\right|$ helps reduce the discrepancy between the learned velocity $v_\theta(t, x_t)$ and the ground truth velocity $u(t, x_t)$. Therefore, the final generation performance can be effectively enhanced under flow matching framework, which is also proved by empirical results in large-scale text-to-image generation (Table 6).
>
> **Novelty and effectiveness of our second term**: Regarding the second term $E_{t, p_t}\left\|v_\theta(t, x_t) - u(t, x_t)\right\|_2^2$ in our final loss function, it is also new to FM methods and is designed to improve training efficiency without significantly degrading model performance. The weight α=1e-5 is empirically chosen based on early experiments because we find that **the first and second terms in the loss function differ by several orders of magnitude, multiplying the second term by 1e-5 can effectively balance the magnitudes of the two terms**. To further enhance the model's expressiveness, we propose to using a multi-segment approach. By dividing the probability flow ODE trajectory into multiple segments, each segment can be more accurately approximated using straight flow.
>
> Based on both theoretical and empirical analysis, our Consistency-FM not only improves generation performance but also enhances training efficiency by combining the two novel terms in final loss function.
>
>
> **Q2: Provide insights into whether these improvements are statistically significant, or if they are due to the flow matching model itself? To clarify this, a baseline using a standard consistency loss (sample space only) in combination with the flow matching model is needed.**
>
> A2: These are really good questions and suggestions. Switching from ODE trajectories derived from diffusion models to flow matching models is indeed one of the reasons for the performance improvement. This improvement is specifically due to the formula we derived from the straight flow theory, which is particularly well-suited for this approach. We conduct an additional experiment on CIFAR-10 dataset, where we use only sample-space consistency loss in combination with the flow matching model. We finally **achieve an FID of 5.29, much better than the FID of 5.83 with consistency model**.
>
> Notably, as shown in Table 6 of our paper, Consistency-FM achieves an FID of 11.02 through one-step generation on the MS COCO 2014 dataset. The Phased Consistency Model uses a standard consistency loss for one-step generation. In comparison, the Phased Consistency Model achieves **an FID of 17.91 (much worse than our FID of 11.02)** in Table 2 of [1] through one-step generation on the same dataset and **with the same backbone**. Thus our improvement over standard consistency model is more significant in large-scale scenarios, demonstraing the effectiveness and superiority of our new FM method.
>
>
> [1] Wang, Fu-Yun, et al. "Phased Consistency Model." NeurIPS 2024.

---

> ### Author Response · Authors · 2024-11-20
> **Response to Reviewer AJPE - Part 2**
>
> **Q3: Illustrating how velocity-space consistency improves model performance compared to traditional sample-space consistency and clarifying the impact of velocity-based consistency on the model’s stability.**
>
> A3: It should be noted that our velocity consistency involves both terms of our loss function, as discussed in our response to Q1. The first term is performed in the sample space with our defined straight flows (for improving both performance and effciency), and the second term directly constrains on velocity consistency (for better trade-off between performance and effciency). As shown in the table below, we conduct additional ablation experiments on the CIFAR-10 dataset.
>
> | Ablation study | Consistency model (traditional sample-space consistency)| Ours ($\alpha$=0)    | Ours ($\alpha$=1e-5) | Ours ($\alpha$=1e-4) |
> | ------------------------------------ |:----: | :----: | :----: | :----: |
> | FID                                  |5.83| 5.29 | 5.34 | 5.43 |
> | Training steps ($\times 10^3$)     | 800|225  | 175  | 145  |
>
> From the results, we can observe that when $\alpha=0$ (using only $\left|f_\theta - f_{\theta^{-}}\right|$), both FID and enficiency can be improved, better than traditional sample-space consistency. With an increase in the weight $\alpha$ for velocity difference ($\left\|v_\theta(t, x_t) - u(t, x_t)\right\|_2^2$), we observed a slight degradation in FID but with the significant enhanced training efficiency.
> This demonstrates that the sample space defined with our velocity consistency (first term) indeed improves both sampling quality and training efficiency while the direct constraint on velocity space (second term) can result in a better trade-off between model performance and convergence.
>
>
>
>
>
>
>
>
>
> **Q4: Provide CIFAR-10 experiments with a distillation sampling variant, the ImageNet 64x64 experiments with a direct consistency training variant. Both the CIFAR-10 and ImageNet experiments omit several key baselines \[2\]\[3\]\[4\].**
>
> A4: Thanks for your suggestions. We provide additional performance comparisons with distillation sampling variant on CIFAR-10 (Table 1) and with direct consistency training variant on ImageNet 64 × 64 (Table 2). We have now included the key baselines [2], [3], [4] in Table 3 and Table 4. We also **update these results in Tab.1 and Tab.2 of our manuscript** which are highlighted in blue for clarity.
>
> Table 1: Performance comparisons with distillation sampling variant on CIFAR-10.
> | Method                        | NFE (↓) | FID (↓)  |
> | ----------------------------- | ------- | -------- |
> | Consistency Distillation      | 2   | 2.93 |
> | CTM (GAN loss)                | 2   | 1.87 |
> | **Consistency-FM (Ours)** | 2       | **1.75** |
>
>
> Table 2: Performance comparisons with direct consistency training variant on ImageNet 64 × 64.
>
> | Method                        | NFE (↓) | FID (↓)  |
> | ----------------------------- | ------- | -------- |
> | Consistency Training             | 2   | 11.10    |
> | **Consistency-FM (Ours)** | 2       | **9.58** |
>
> Table 3: Performance comparisons on CIFAR-10.
>
> | Method                        | NFE (↓) | FID (↓)  |
> | ----------------------------- | ------- | -------- |
> | iCT-deep [3]                  | 2   | 2.24 |
> | TRACT [2]                     | 2   | 3.32 |
> | MultiStep-CT [4]              | 2   | 2.87    |
> | **Consistency-FM (Ours)** | 2       | **1.75** |
>
> Table 4: Performance comparisons on ImageNet 64 × 64.
>
> | Model                                         | NFE   | FID(↓)   |
> | ------------------- | ----- | -------- |
> | PD                                            | 2     | 8.95     |
> | TRACT [2]                                     | 2 | 4.97 |
> | CD                                            | 2     | 4.70     |
> | iCT-deep [3]                                  | 2 | 2.77 |
> | MultiStep-CD [4]                              | 2 | 1.9  |
> | **Consistency-FM (Ours)**                 | 2     | **1.62** |
>
>
> [2] Berthelot, D., Autef, A., Lin, J., Yap, D.A., Zhai, S., Hu, S., Zheng, D., Talbott, W. and Gu, E., 2023. Tract: Denoising diffusion models with transitive closure time-distillation. *arXiv preprint arXiv:2303.04248*.
>
> [3] Song, Y. and Dhariwal, P., 2023. Improved techniques for training consistency models. *arXiv preprint arXiv:2310.14189*.
>
> [4] Heek, J., Hoogeboom, E. and Salimans, T., 2024. Multistep consistency models. *arXiv preprint arXiv:2403.06807*.
>
> **Q5: Include an analysis of how sample quality changes with different NFE steps, especially for the multi-segment Consistency-FM model.**
>
> A5: As shown in the table below, sample quality improves with an increase in the number of evaluations (NFE) for the multi-segment Consistency-FM model on CIFAR-10.
>
> | NFE  | 2    | 4    | 6    |
> | ---- | ---- | ---- | ---- |
> | FID  | 5.34 | 3.97 | 3.04 |
>
> This demonstrates that increasing NFE steps enhances the quality of generated samples, validating the effectiveness of the multi-segment approach in improving model performance.

---

> ### Author Response · Authors · 2024-11-25
> **Gentle Reminder**
>
> Dear reviewer AJPE:
>
> We sincerely appreciate the time and effort you dedicated to reviewing our paper. In response to your concerns, we have conducted additional experiments and provided an in-depth analysis to demonstrate both effectiveness and efficiency of our proposed Consistency-FM during the discussion period.
>
> As the discussion period concludes in two days, we kindly request, if possible, that you review our rebuttal at your convenience. Should there be any further points requiring clarification or improvement, please know that we are fully committed to addressing them promptly. Thank you once again for your invaluable contribution to our research.
>
> Warm regards,
>
> The Authors

---

> ### Comment · Reviewer_AJPE · 2024-11-26
>
> Thank you to the authors for their additional experiments and discussions. While these efforts address some of my concerns, my primary issue regarding the effectiveness of the additional velocity-space consistency remains unresolved, which directly affects the perceived novelty of this work. Specifically, the provided sample-space-only experiment achieves an FID score of 5.29, outperforming all configurations that utilize velocity-space consistency with a positive $\alpha$. The argument for training efficiency is also unpersuasive, as one could similarly argue that fewer training iterations with sample-space-only consistency loss could yield comparable FID scores.
>
> Regarding the novelty of the sample-space consistency loss itself for flow-matching models, its contribution appears limited. Rectified flow is a specific case within the broader class of diffusion models, with the forward process sharing a formulation akin to $x_t = a_t x_0 + b_t \epsilon$ [2]. Additionally, velocity prediction in flow models and noise prediction in diffusion models are interchangeable [2, Eq. 12]. As a result, the proposed sample-space loss function closely resembles the original consistency function introduced in Consistency Models [1, Eq. 7 and Eq. 9]. Substituting $f_\theta$ from the consistency model for diffusion models with flow-matching models, as described in [2, Eq. 13 and Eq. 14], is a straightforward extension. Thus, the novelty of this paper seems limited, and I am inclined to maintain my current score.
>
> [1] Song, Y., Dhariwal, P., Chen, M., & Sutskever, I. (2023, July). Consistency Models. *International Conference on Machine Learning*, 32211–32252. PMLR.
>
> [2] Esser, P., Kulal, S., Blattmann, A., Entezari, R., Müller, J., Saini, H., Levi, Y., Lorenz, D., Sauer, A., Boesel, F., & Podell, D. (2024, March). Scaling Rectified Flow Transformers for High-Resolution Image Synthesis. *Forty-First International Conference on Machine Learning*.

---

> > ### Author Response · Authors · 2024-11-26
> > **Response to Reviewer AJPE**
> >
> > Thank you for your feedback. In response to your remaining concerns, we further make clarification of these two questions:
> >
> > 1. Please kindly note that our provided sample-space-only experiment is based on the first term of our loss function, and **this sample space is defined by our straight flows utilizing velocity-space consistency**, which is different from traditional sample-space. Thus, the effectiveness is well demonstrated. And our training efficiency means that our model can achieve better performance with same training iterations. Fewer training iterations with sample-space-only consistency loss can only yield much worse FID scores.
> >
> > 2. It is noted that our **Consistency-FM defines straight flows with velocity consistency that innovatively bridges consistency models and flow matching models** in a simple yet effective manner. Such design is new and effective as demonstrated by both qualitative and quantitative results in our paper. We also provide ablation study and explanations in the rebuttal to prove the uniqueness of our Consistency-FM.
> >
> > We hope our response helps you better understand our method and addresses your concerns. If you have any further questions, please feel free to reach out for further discussion.

---

### Author Response · Authors · 2024-11-20
**Global Response**

We sincerely thank all the reviewers for their thorough reviews and valuable feedback. We are glad to hear that  our proposed framework is novel and interesting (all reviewers), theoretical analysis is detailed and sound (reviewer 4ccL and Yzev) and our multi-segment Consistency-FM model is novel and effective (reviewer AJPE and Yzev). We have revised the manuscript according to the suggestions of reviewers (**marked in blue**).


We summarize our responses to the reviewers' comments as follows:

* We additionally provide more comparative results to demonstrate the superior generation quality and training efficiency of our framework.

* We provide more theoretical and empirical analysis to clarify the critical contributions of the two terms in our loss function.

* We conduct more ablation studies and provide more experimental details for better understanding of our framework.

We reply to each reviewer's questions in detail below their reviews. Please kindly check out them. Thank you and please feel free to ask any further questions.

---

### Meta-Review · Area_Chair_GR36 · 2024-12-23

**Metareview:**

The paper proposes Consistency-FM, a novel approach integrating velocity-space consistency into flow matching models to enforce straight flows. This design aims to improve both training efficiency and generation quality, with a multi-segment training strategy enhancing expressiveness. Experimental results indicate significant improvements over baselines in training convergence and FID scores across several datasets.

The reviewers largely agree on the method's novelty and its potential contributions to generative modeling. Strengths include its theoretical underpinnings, empirical performance, and detailed experimental validation. However, some weaknesses were noted: marginal improvements in certain metrics, limited baseline comparisons in initial results, and questions regarding the necessity of velocity-space consistency. While reviewers AJPE and Yzev initially raised concerns, most were addressed satisfactorily during rebuttal, except for AJPE's reservations about the effectiveness of the velocity consistency component.

The AC considers that the paper's contributions to flow matching methodologies are substantial, and its performance improvements warrant dissemination despite minor unresolved concerns. The paper's quality by itself marginally passes the bar to be accepted at ICLR. However, the supplementary file contains a technical report of this work with author names (github_misc->Consistency_FM_arxiv.pdf), which is a violation of the ICLR double blind submission policy (https://iclr.cc/Conferences/2025/AuthorGuide) that "supplementary material will be visible to reviewers and the public throughout and after the review period, and ensure all material is anonymized." Considering this, the AC recommends rejection at this point.

**Additional Comments On Reviewer Discussion:**

Please refer to the above metareview.

---

### Decision · Program_Chairs · 2025-01-22

Reject